



# Can mussels be used as sentinel organisms for characterisation of pollution in urban water systems?

**Elke S. Reichwaldt[1] and Anas Ghadouani[1]**

[1] Aquatic Ecology and Ecosystem Studies, School of Civil, Environmental and Mining Engineering, M015, The University of Western Australia, 35 Stirling Highway, Crawley, Western Australia 6009, Australia.

Correspondence to: E.S. Reichwaldt (elke.reichwaldt@uwa.edu.au)

## Abstract

Urbanisation strongly impacts aquatic ecosystems by decreasing water quality and altering water cycles. Today, much effort is put towards the restoration and conservation of urban waterbodies to enhance ecosystem service provision leading to liveable and sustainable cities. To enable a sustainable management of waterbodies, the quantification of the temporal and spatial variability of pollution levels and biogeochemical processes is essential. Stable isotopes have widely been used to identify sources of pollution in ecosystems. For example, increased nitrogen levels in waterbodies are often accompanied with a higher nitrogen stable isotope signature ($\delta^{15}$N), which can then be detected in higher trophic levels such as mussels. The main aim of this study was to assess the suitability of nitrogen stable isotope as measured in mussels, as an indicator able to resolve spatial and temporal variability of nutrient pollution in an urban, tidally influenced estuary (Swan River estuary; Western Australia). Our results showed a trend by which sites with higher nitrates concentrations yielded higher nitrate $\delta^{15}$N values; however, nitrogen concentrations and nitrogen stable isotope signature of nitrate throughout the estuary were well within natural values, indicating groundwater inflow rather than pollution by human activity was responsible for differences between sites. The $\delta^{15}$N signature in mussels was very stable over time within each site which allowed for the detection of spatial difference and indicated that mussels can be used as time-integrated sentinel organism in urban systems. In addition, our study indicates that the nature of the relationship between $\delta^{15}$N



in the mussels and the nitrate in the water can provide insights into site specific
biogeochemical transformation of nutrients. We suggest that mussels and other sentinel
organisms can become a robust tool for the detection and characterization of the dynamics
of a number of emerging anthropogenic pollutants of concern in urban water systems.
## 1   Introduction
Humans exert a growing impact on the environment supporting them. Today, more than
50% of the world's population is living in cities and this percentage is projected to further
increase to up to 80% by 2050 (Pickett et al., 2011; United Nations, 2013). The high
percentage of impervious surfaces and the high population density in cities lead to drastic
changes in the water cycle and water quality in a range of urban water systems, including
lakes, wetlands, rivers, streams, estuaries and coastal ecosystems. Impervious surfaces lead
to less rainfall infiltrating the soil. Instead, stormwater runoff is directly transported to
waterbodies, polluting them with nutrients, heavy metals, and bacteria (Makepeace et al.,
1995; Brezonik and Stadelmann, 2002). Urbanisation has resulted in increased
eutrophication of waterbodies leading to deteriorated ecosystems worldwide, reducing
natural biodiversity and ecosystem services (Heathwaite, 2010). In an attempt to reconnect
cities to their natural water resources, much effort is going not only towards the restoration
and conservation of existing waterbodies, but also to increasing our understanding on how
to manage those ecosystems that are irreversibly altered by man, sometimes referred to as
"novel ecosystems" or never-before-seen ecosystems (Hobbs et al., 2014; Collier, 2015).
The greater need for a full integration between the management and restoration of existing
ecosystems and the introduction and interventions of new ecosystems is especially needed
as statutory planning for cities of the future puts greater emphasis on the provision of a
wide range of ecosystem services and its full integration in the landscape (Plieninger et al.,

26    2014).

Typically the success rate of restoring degraded waterbodies is highly variable
(Søndergaard et al., 2007) and it is anticipated that the management of ecosystems in the
urban environment will emerge even more challenging given the added complexities
discussed above. Environmental management is often hampered by a limited
understanding of the temporal and spatial variability of pollution levels, the sources of
contamination and the processes within systems that affect the recovery of a system



(Kooistra et al., 2001; Scheffer et al., 2001; Lahr and Kooistra, 2010). In addition, the
traditional hierarchical water management practices that are still in use around the world
have been criticised as being ineffective and leaving little scope for adaptation to changes
(Pahl-Wostl, 2007; van de Meene et al., 2011). The current trend to decentralise urban
water management might allow for more local management of water resources, indicating
the need for improving our understanding of the variability of pollution levels in a range of
urban waterbodies with greater emphasis on local processes.
Many urban estuaries are highly impacted by human activity due to direct input of
pollutants from urban, agriculture and industry areas (e.g., Oczkowski et al., 2008) and
will be even more impacted in the future. Nutrient pollution is of particular concern in
many waterbodies, because it can lead to eutrophication. In urbans estuaries, tributaries
often transport high amounts of nutrients from the watershed into the estuary, causing
water quality problems including toxic bloom development (Hamilton, 2000; Atkins et al.,
2001). Nutrient concentration gradients might develop with higher upstream and lower
downstream values, where pollution is diluted by seawater (Dähnke et al., 2010).  This can
lead to a spatial variability of nutrient concentration within estuaries. Nutrient pollution
can also be highly variable in time with higher nutrient concentrations in estuaries found
during times of high water input by tributaries. Smaller scale variability in temporal and
spatial nutrient concentrations can additionally stem from local differences in hydrological
processes (Linderfelt and Turner, 2001) and variations in fertilizer use in agricultural areas
or temporal failure of septic tank systems leading to leakage of sewage, leading to
localised places of concern for water management.
Anthropogenic nutrient and organic pollution of water systems, including the interaction
between surface and groundwater, have been successfully investigated using a range of
stable isotopes (Sikdar and Sahu, 2009; Yang et al., 2012; Lutz et al., 2013). In addition,
stable isotopes have been widely used in purely hydrological studies focused on flow
paths, hydraulic residence time and other hydrological dynamics (Clay et al., 2004;
Rodgers et al., 2005; Volkmann and Weiler, 2014). Stable isotopes of nitrogen (N), carbon
(C), sulfur (S) and oxygen (O) in water and biota have also been applied as an integrated
measure of ecosystem processes (Robinson, 2001; Chaves et al., 2003; Pace et al., 2004).
Furthermore, the   analysis of the nitrogen signature has proven to be an especially
powerful tool as an indicator of anthropogenic contamination (Lake et al., 2001;
McKinney et al., 2002; Fry and Allen, 2003) and landuse (Harrington et al., 1998;





Broderius, 2013; Carvalho et al., 2015), bearing on the fact that the sources of
contamination such as animal manure, sewage, septic waste, some fertilizers carry higher
nitrogen signatures values and consequently a higher $\delta^{15}N$ (Heaton, 1986; Cabana and
Rasmussen, 1996; Kellman, 2005; Choi et al., 2007). This signal is then passed on to
higher trophic levels up the food chain (e.g., Cabana and Rasmussen, 1994; Harrington et
al., 1998; Carvalho et al., 2015).
Assessing anthropogenic pollution of a system by directly measuring the isotopic signature
of nitrogen containing nutrients (e.g., nitrate, ammonium) or of aquatic short-lived
organisms with fast tissue turnover times, such as phytoplankton, may significantly under-
or overestimate the average level of pollution, as the result strongly depends on the time of
measurement. Mussels on the other hand, which are primary consumers with limited
movement, have been suggested as suitable site-specific bioindicators of time-averaged
persistence of nutrient pollutants, because their isotopic signature fluctuates less than that
of their food sources due to longer tissue turnover rates (Raikow and Hamilton, 2001; Post,
2002; Fukumori et al., 2008; Fertig et al., 2010). Earlier studies in polluted freshwater and
marine systems found positive relationships between the concentration of nitrogen and the
isotopic signature of nitrogen in mussels, and between the isotopic signature of nitrate-N
and that of mussels. This suggests that bivalves are suitable indicators of changes in
nutrient pollution load to waterbodies (Cabana and Rasmussen, 1996; McClelland et al.,
1997; Costanzo et al., 2001; Anderson and Cabana, 2005; Gustafson et al., 2007; Wen et
al., 2010). However, very little information exists on the use of these stable isotopic
signatures in urban systems.
The main aim of this study was to identify the variability of nitrogen concentration in an
urban estuary over time and space and to ascertain the suitability of the isotopic signature
($\delta^{15}N$) of mussel tissue as an indicator of nitrogen pollution in urban water systems.
Specifically, we anticipated that (1) a higher input of nitrogen rich waters upstream would
lead to a higher isotopic signatures, (2) distinct spatial difference in mussels are driven by
the level of nitrates in the water, and (3) the increased distance from the mouth would lead
to an increased anthropogenic signal in the mussels due to the freshwater input.

## 2  Materials and Methods

### 2.1  Study sites





The study was performed in the lower reaches of the heavily urbanised Swan River estuary
that flows through Perth, Western Australia (Fig. 1) (Atkins and Klemm, 1987). The
catchment of this estuary is approximately 121,000 km$^2$ (Peters and Donohue, 2001) and
encompasses urban, rural, agricultural and forested areas. In the urban area, drains contain
sewered and unsewered areas (Peters and Donohue, 2001). The Swan River estuary
experienced a major toxic cyanobacterial bloom in 2000, when a large rainfall event
increased nutrient concentrations and decreased salinity within the estuary (Hamilton,
2000; Atkins et al., 2001), indicating that this estuary is prone to pollution from the
watershed. The Swan River estuary is influenced by mostly diurnal tides with a mean tidal
range at the mouth of the estuary of 0.8 m. At the same time, the estuary is seasonally
forced with a large discharge of freshwater from the tributaries during the wetter winter
months (May to September), and little freshwater discharge during dry summers. This
leads to fresh to brackish water in parts of the estuary in winter with a freshwater lens
overlying saltwater, and an inland progression of the saltwater wedge, making the estuary
a saltwater habitat during drier months (Stephens and Imberger, 1996). The Swan River
estuary is permanently open to the ocean and has two major freshwater tributaries, the
Swan River and the Canning River (Fig. 1). While there are also several short stormwater
drains leading into the lower Swan River estuary that could potentially provide nutrient
input into the Swan River estuary from the adjacent land, these drains did not flow during
the study.
Seven sites within the Lower Swan River estuary were sampled 6 times for mussels and 9
times for nutrients, chlorophyll-$a$, temperature, salinity, pH and oxygen during the wetter
season (March - November 2010). The sites were jetties at Point Walter (WP) (32° 0'
39.23" S, 115° 47' 15.11" E), Minim Cove Park (MC) (32° 1' 21.23" S, 115° 45' 57.38" E),
Swan River Canoe Club (SCC) (32° 0' 27.31" S, 115° 46' 18.73" E), Claremont (Cl) (31°
59' 23.80" S, 115° 46' 52.97" E), Broadway (BRD) (31° 59' 25.55" S, 115° 49' 5.49" E),
Applecross (AC) (32° 0' 17.59" S, 115° 49' 58.29" E), Como Beach (CB) (31° 59' 37.46"
S, 115° 51' 10.33" E) (Fig. 1). While MC and SCC are situated at the deeper part of the
estuary (depth < 17 m), all other sites are located in the shallower part (depth < 10 m)
(Stephens and Imberger, 1996). The jetty at Cl is situated in a shallow bay (depth
approximately 2 m) with established seagrass meadows and abundant macroalgae and
macrophytes (Department of Water, 2010). Additionally, a one-time marine reference





measurement was performed towards the end of the study outside the estuary at Woodman
Point Jetty (WO; 32° 7' 26.97" S, 115° 45' 32.10" E) (Fig. 1).
## 2.2 Sampling and analyses
On each date, sampling was performed 0.5 to 1 h prior to high and low tide at each site,
respectively. While mussels were sampled only once per day, all other parameters were
sampled at high and low tide. Salinity, pH, water temperature and oxygen were measured
at 20 cm depth with hand-held probes (WP-81; TPS-DO$_2$). At each site, one water sample
for quantification of nutrient concentration (TP = total phosphorous, NO$_x$ = nitrate (NO$_3$) +
nitrite (NO$_2$), NH$_4^+$ = ammonium), phytoplankton biomass (as chlorophyll-*a*), and stable
isotope analysis of NO$_3$ ($\delta^{15}$N, $\delta^{18}$O) and particulate organic matter (POM; $\delta^{15}$N) were
taken from 10 to 20 cm below the surface and brought back to the laboratory in glass
bottles that were stored on ice. Nine mussels per site were randomly taken from the pylons
of the jetties at each site from between 20 and 40 cm depth and brought into the laboratory
on ice in bags containing water from the respective site. There were no mussels at WP in
November.
In the laboratory, total phytoplankton concentration at each site was measured with a
bench top version of the FluoroProbe (bbe Moldaenke, Germany) as μg chl-*a* L$^{-1}$ (Beutler
et al., 2002; Ghadouani and Smith, 2005). Water for quantification of NO$_x$ (LOQ = 0.14
μM) and NH$_4^+$ (LOQ = 0.21 μM) concentrations was filtered through 0.45 μm syringe
filters (Ht Tuffryn, Pall, Australia) and kept frozen until analysis at the Marine and
Freshwater Research Laboratory (Murdoch University, Western Australia) using a Lachat
Quikchem Flow Injection Analyser. Water for analysis of nitrate $\delta^{15}$N was filtered through
0.2 μm syringe filters (Ht Tuffryn, Pall, Australia) and kept frozen until analysis at the UC
Davis Stable Isotope Facility (Davis, California, USA) using a ThermoFinnigan GasBench
plus PreCon trace gas concentration system interfaced to a ThermoScientific Delta V Plus
isotope-ratio mass spectrometer (Bremen, Germany), with the bacteria denitrification
method (Sigman et al., 2001). The limit of quantification for this analysis was 0.71 μM
NO$_3$-N and the external errors of analysis were 0.4 ‰ for nitrate $\delta^{15}$N and 0.8 ‰ for
nitrate $\delta^{18}$O. Raw water was used for quantification of TP with the ascorbic acid method
(APHA, 1998).





For analysis of nitrogen stable isotope signature of particulate organic matter (POM) as the
food for mussels, 0.7 - 2.5 L of water was filtered onto pre-combusted 25 mm GF/C filters
(Whatman), which were then dried for 24 h at 60°C and stored in a desiccator until
analysis. After determining mussel length to the nearest millimetre they were dissected to
obtain the foot tissue for stable isotope analysis. The feet of three individuals per site were
combined, dried at 60°C for at least 24 h and stored in a desiccator until analysis for
mussel $\delta^{15}N$ and C:N ratio. As 9 mussels per site were collected, this resulted in three
replicates for stable isotope analysis per site, each replicate comprised of the feet of three
mussels. This method was adopted from Lancaster and Waldron (2001) as the minimum
detectable difference between two populations was negatively associated with the number
of replicate samples and the number of individual animals combined in each replicate.
Therefore, this method is preferred, when only small differences in the stable isotope
signatures are expected. We used foot tissue for the analysis, because it is easy to identify
and obtain, and because its $\delta^{15}N$ value presents a time-averaged value of $\delta^{15}N$ of the food
source. Stable isotope analysis of mussel feet tissue and POM was performed at the West
Australian Biogeochemistry Centre (University of Western Australia, Australia) with a
continuous flow Delta V Plus mass spectrometer (connected with a Thermo Flush 1112 via
Conflo IV) (Thermo-Finnigan, Germany). The external errors of analysis were 0.10 ‰ for
$\delta^{15}N$. To check whether the size of mussels was correlated with their $\delta^{15}N$, 13 mussels with
shell lengths between 30 and 54 mm were sampled from MC in July.
**2.3   Data processing and statistical analyses**
Relationships between parameters (i.e. nutrient concentrations, physical parameters, chl-*a*,
stable isotope values) and distance to the estuary mouth were analysed with linear
regressions. Differences between sites were analysed with one-way ANOVA or Kruskal
Wallis one-way ANOVA, in cases where the normality test failed (Sokal and Rohlf, 1995).
If significant, the parametric Tukey (equal variances) or the non-parametric Games Howell
(non-equal variances) post hoc tests were used to identify which sites were different. The
Mann-Whitney U test was used to compare chl-*a* concentrations between high and low
tide. All analyses were done with IBM® SPSS® Statistics 20 or Sigma Plot® Statistics 11.0,
and significance level was set to $p < 0.05$ unless stated otherwise.





## 3 Results
## 3.1 Physicochemical parameters
Rainfall was below average in 2010 with 416 mm for the entire sampling period, while the
long-term average for this period is 677 mm. This resulted in a lower than usual discharge
from the tributaries into the estuary with a mean discharge from the Swan River of
$1.2 \times 10^5$ m$^3$ d$^{-1}$ in 2010 (Water Information System, Department of Water, Western
Australia) compared to $1.4 \times 10^6$ m$^3$ d$^{-1}$ in 1993-1994 for the same season (Hamilton et al.,
2006). This might have contributed to unseasonally high salinities throughout the entire
estuary during this study and no relationship between salinity and distance to the estuary
mouth was detected. During high tide, the salinity at all sites was between 24.2 and 32.4
and there was no difference in salinity between sites. Although salinity was not different
between sites at low tide either, sites further away from the ocean (AC, CB, BRD) were
entirely freshwater between March and June, while saline (mean ± SE; 27.4 ± 0.4)
conditions prevailed at all sites between July and November. There were no differences
between sites in temperature (temporal range 12.5 – 23°C; Kruskal-Wallis H = 0.584,
df = 6), dissolved oxygen (temporal range 6.4 - 11.6 mg L$^{-1}$, one-way ANOVA
$F_{6,84} = 0.764$; 63 – 124 % *sat.,* one-way ANOVA $F_{6,84} = 0.515$), and pH (temporal range
6.7 - 8.4; one-way ANOVA $F_{6,112} = 0.163$). Total chl-*a* concentration was between 1.4 and
9.5 µg L$^{-1}$ with a mean of 3.9 µg L$^{-1}$ (CV = 0.18). Total chl-*a* concentration was similar
between sites (ANOVA; $F_{6,70} = 1.45$), and did not differ between low and high tide at any
site (Mann-Whitney U Test).
## 3.2 Nutrient concentrations
Overall, NO$_x$ and NH$_4^+$ concentrations were low in the Swan River estuary. The
concentration of NO$_x$ ranged between below quantifiable limits (LOQ = 0.14 µM) and
15.0 µM (median 0.29; mean ± SD 0.72 ± 1.7), and differed significantly between sites
(Kruskal Wallis One way ANOVA, H = 50.03, df = 6) (Fig. 2). The concentration of NH$_4^+$
ranged between the limit of quantification (LOQ = 0.21 µM) and 2.6 µM (median 0.78;
mean ± SD = 0.85 ± 0.58) and did not differ between sites (Kruskal Wallis One way
ANOVA, H = 7.9, df = 6). On average, NO$_x$ was the dominant N source at MC, SCC and
WO, while it was NH$_4^+$ at all other sites (Fig. 2). This is supported by the significant
difference in the mean fraction of NO$_x$ of total dissolved nitrogen between sites (Kruskal





Wallis one-way ANOVA, H = 59.0, df = 6) with site MC having a higher fraction than all
other sites and sites SCC and WP being intermediate (data not shown). Total phosphorous
was below or just above the limit of quantification (LOQ = 0.32 µM) throughout the study
and did not show any spatial or temporal trend. The TN:TP ratio (weight) was between 0
and 6.5. Traditionally nitrogen limitation was said to occur at ratios (weight) below 7.2
(Redfield, 1958), however, more recent work indicated that the TN:TP ratio (weight) of
marine matter and nutrient-replete phytoplankton can range from 2.2 to 15.4 (Geider and
La Roche, 2002), suggesting that the ratio of 7.2 might be too high. In our experiment 84%
of the ratios were below 2.2, indicating a high possibility of nitrogen limitation in this
system.
The concentrations of total dissolved inorganic nitrogen (TDIN = $NO_x$ + $NH_4$) (µM) and
$NO_x$ (µM) were higher towards the estuary mouth (Fig. 2), although these relationships
were weak (TDIN: $r^2$ = 0.113, y = -0.186x+3.69, $F_{1,117}$ = 14.86; $NO_x$: $r^2$ = 0.153, y = -
0.196x+2.98, $F_{1,117}$ = 21.16) and were driven by site MC only. Ammonium concentrations
were not correlated with the distance from the estuary mouth ($F_{1,117}$ = 0.41).
**3.3    Stable isotope values of $NO_3$**
Analysis of the stable isotope signature of $NO_3$ was limited to a total of 25 samples that
fulfilled nutrient concentration requirements for the analysis. Of these, 9 were from MC,
10 from SCC, 2 from AC, 3 from CB, and 1 from WP. Nitrate $\delta^{15}N$ values varied between
-1.3 and 10.4 ‰, while nitrate $\delta^{18}O$ values ranged between 18.4 and 72.9 ‰. Nitrate $\delta^{15}N$
differed between sites (one-way ANOVA; $F_{4,25}$ = 5.94) and increased exponentially with
increasing $NO_x$ concentration ($F_{1,23}$ = 10.50) (Fig. 3). A post-hoc test (Games Howell)
indicated that nitrate at MC was $^{15}N$ enriched (mean ± SD; 7.92 ‰ ± 2.55; n = 12)
compared to SCC (2.71 ‰ ± 1.02; n = 10) and AC (-0.19 ‰ ± 1.51; n = 2). There was no
temporal trend in nitrate $\delta^{15}N$ at sites MC and SCC, respectively, which were the only two
sites for which sufficient data for such an analysis were available. Nitrate $\delta^{18}O$ was not
significantly different between sites ($F_{4,25}$= 0.059).

**3.4    Stable isotope values of POM**
POM $\delta^{15}N$ values were between 6.2 and 9.9 ‰ with no significant difference between sites
($F_{6,25}$= 1.327). A significant positive relationship between nitrogen stable isotope





signatures of POM and mussels was found ($r^2$ =0.303, y = 0.20x + 7.40, $F_{1,14}$ = 6.08), with
an average fractionation of 0.6 ‰.

## 3.5 $\Delta^{15}$N of mussels

No significant relationship between mussel length and mussel $\delta^{15}$N (linear regression;
$F_{1,13}$ = 2.235) was found. Values of $\delta^{15}$N of mussels varied between 6.8 and 10.3 ‰ and
the range was therefore smaller than the range seen in nitrate $\delta^{15}$N. No temporal trend in
mussel $\delta^{15}$N was detected (Fig. 4). $\Delta^{15}$N of mussels was significantly different between
sites (one-way ANOVA; $\delta^{15}$N: $F_{6,98}$ = 42.53) (Fig. 5) and mussel $\delta^{15}$N increased with
increasing distance from the estuary mouth (Fig. 6).
Mussel $\delta^{15}$N was negatively correlated with the concentration of total dissolved inorganic
nitrogen ($r^2$ = 0.486, $F_{1,5}$ = 4.73, $P < 0.1$) (Fig. 5). When site Cl was omitted, the strength
of the relationship increased ($r^2$ = 0.838, $F_{1,4}$ = 20.69, $P < 0.05$), while the relationship was
not significant with an $r^2$ of 0.009 only when sites MC was omitted (Fig. 5). There was a
significant negative relationship between the $\delta^{15}$N values of mussel and nitrate (Fig. 7) ($r^2$
= 0.711, $F_{2,10}$= 24.65).

## 4 Discussion

Urban development poses a major threat to aquatic ecosystems, resulting in a range of
systems with different impact levels. The management of these waterbodies, whether they
are historical, hybrid or novel (Hobbs et al., 2014), requires a detailed knowledge on the
complex interactions of processes in these systems. The limited understanding of spatial
and temporal variabilities of pollutants is often the major limitation to successful and long-
lasting restoration and protection efforts (Kooistra et al., 2001; Lahr and Kooistra, 2010).
As such it is essential to develop in-depth knowledge of local processes and pollution
levels that will allow a decentralised management approach adapted to local issues (van de
Meene et al., 2011).
Our study supports this notion by showing that the concentration of nitrates and the
nitrogen stable isotope signatures of nitrate and of mussels were different between sites in
the Swan River estuary. Site-specific differences in nutrient concentrations can be caused
by local input of nutrients or by site-specific differences in nutrient cycling caused by



physicochemical conditions or biological factors (Michener and Lajtha, 2007).
Additionally, nutrient input from the watershed often leads to higher nutrient
concentrations upstream. During our study, freshwater input into the estuary was weak,
leading to the estuary being mainly influenced by ocean water. This might have been the
reason that no increase of nutrients upstream was found in this study and that nitrogen
concentrations were in general low. However, differences in $NO_x$ and TDIN
concentrations between sites suggested a significant site-specific input of nutrients into the
Swan River estuary. This is supported by the fact that mean nitrogen concentrations at the
site closest to the ocean (MC) were higher than the concentrations in the ocean (WO)
pointing towards a local input of non-marine $NO_x$ at MC.
Earlier studies indicated that the nitrogen stable isotope ratio of dissolved inorganic
nitrogen was often higher at sites with high anthropogenic nitrogen pollution (Heaton,
1986; Cabana and Rasmussen, 1996). In the Swan River estuary, $NO_3$ was enriched and
there was a positive relationship between nitrate $\delta^{15}N$ and the concentration of $NO_x$
throughout the estuary. However, because the isotopic signatures of nitrates were well in
the range of values reported for surface water, uncontaminated groundwater (Xue et al.,
2009), or organic nitrate from soils (Heaton, 1986), our study does not suggest differences
in the level of human impact between sites. Additionally, nitrate $\delta^{18}O$ values are similar to
values indicative of the atmospheric source (Kendall, 1998; Xue et al., 2009), suggesting
that the higher concentration and enriched signature of $NO_x$ at site MC is unlikely to result
from anthropogenic pollution, but might rather be due to addition of $NO_x$ by groundwater
inflow, potentially in combination with different productivity or biochemical processes at
this site compared to any of the other sites.
Part of the site specific variation in nitrate $\delta^{15}N$ in this study can be explained by the
fraction of $NO_x$ of the TDIN pool (%) (data not shown; $y = 0.15x-6.9$, $r^2 = 0.215$, $F_{1,23} =$
6.30, $P < 0.05$). This is similar to what Sugimoto et al. (2009) found in their study in a
eutrophic coastal environment and which they explained by *in situ* isotopic effects during
nitrification. However, ammonium concentrations in our system were below 5 µM, so that
nitrification in the water column was unlikely to play a major role (Day et al., 1989). This
is further supported by the high $\delta^{18}O$ values of nitrate in our system which is, together with
the $\delta^{15}N$ signature of $NO_3$ rather representative of atmospheric $NO_3$ deposition values
(Durka et al., 1994; Fang et al., 2011).





Nitrogen $\delta^{15}N$ values are reflected in higher trophic levels in a predictable way with
primary consumers (e.g., mussels) from sites with higher nitrate $\delta^{15}N$ values also having
higher $\delta^{15}N$ values (Cabana and Rasmussen, 1996; Oczkowski et al., 2008). Earlier studies
have also shown a positive relationship between primary producer and primary consumer
$\delta^{15}N$ values (Cabana et al., 1994; Harrington et al., 1998; Carvalho et al., 2015). Our study
showed a positive relationship between food (POM) and mussel $\delta^{15}N$, but a negative
relationship between nitrate $\delta^{15}N$ and consumers (mussels). Such negative relationships
were previously found in systems with very high nitrogen concentrations (DIN > 40 μM)
(Oczkowski et al., 2008), because in these systems primary producers can be choosy and
will preferentially uptake lighter $NO_x$, leading to a higher fractionation at higher
concentrations (Lake et al., 2001; Oczkowski et al., 2008). Therefore, the residual $NO_x$ in
those waters retains more $^{15}N$-enriched material, leading to a positive relationship between
nitrogen concentration and nitrate $\delta^{15}N$, while consumers which incorporate primary
producers will have a lighter signature. Because such fractionation is unlikely at TDIN
concentrations below 1 μM (Oczkowski et al., 2008), this mechanism is unlikely for most
of our sites where mean TDIN concentration was < 1.5 μM. This is also supported by the
lack of relationship between mussel $\delta^{15}N$ and TDIN concentration when omitting MC.
However, we cannot rule out that this mechanism partially contributed to the low mussel
$\delta^{15}N$ values detected at MC as TDIN concentrations were higher at this site with a mean of
3.6 μM.
The relationship between mussel $\delta^{15}N$ and TDIN concentration was much higher when
omitting site Cl. This site was the shallowest site with a high density of macroalgae and
seagrass. These benthic primary producers are known to incorporate nutrients from the
groundwater and pore water (Pennifold and Davis, 2001). As pore water in the Swan River
estuary contains a high concentration of ammonium (Linderfelt and Turner, 2001), this is
taken up by the benthic primary producers, and, when recycled, nitrogen with a different
$\delta^{15}N$ value is released into the water column. Therefore, nitrogen $\delta^{15}N$ in the water column
at this site is likely to differ from that of all other sites, which could explain why mussel
$\delta^{15}N$ values at Cl do not fit the general negative relationship. Due to constantly low nitrate
concentration at this site, the stable isotope signature of nitrate could not be tested in our
study.
Fluctuation of mussel $\delta^{15}N$ at each site over time was low compared to the differences
between sites, indicating that observed differences between sites prevailed and were not





obscured by time effects. This is important for assessing site-specific source inputs. The
limited temporal variation likely reflected the physiochemical state of the system during
the study period; in our study, the estuary was dominated by marine influences due to
reduced river discharge. This might have further resulted in a dampening effect of possible
fluctuations of the nitrate $\delta^{15}N$ value caused by changes in watershed input. Our results
therefore indicate that while high seasonal variations of stable isotope signature in mussels
can be connected to seasonal changes in watershed input and chemistry in large rivers (Fry
and Allen, 2003), this is less pronounced in tidally influenced estuaries.
We found an increase in the nitrogen stable isotope signal in the mussels with increasing
distance from the estuary mouth. This contrasts an earlier study in a heavily polluted
estuary showed only little spatial variability ($< 0.4$ ‰) of clam $\delta^{15}N$ values between
upstream (polluted) sites and sites close to the mouth (unpolluted) of the estuary
(Oczkowski et al., 2008). They argued that all clams within their system relied to a large
portion on phytoplankton that used upstream nitrogen sources. In our study, differences in
mussel $\delta^{15}N$ values between sites were larger ($<1.3$ ‰) than in their study, and stable,
although we did not find very large differences in nitrogen concentration or nitrate $\delta^{15}N$
values. Differences in mussel $\delta^{15}N$ values between sites in our study could be due to the
fact that mussels rely on local primary production, which in turn might depend on site
specific nitrogen sources such as nitrate and ammonium. As nitrate and ammonium were
found to be taken up with different isotopic fractionation by primary producers (Pennock
et al., 1996), this would then be reflected in the mussels.
**5   Conclusion**
The findings of our study corroborate that stable isotope analysis is a valuable tool for
identifying spatial variability of nutrient pollution and local processes in an urban, tidally
influenced estuary. As such, stable isotope analysis can deliver essential information for
future decentralised water management practices that are focused on local process
understanding. We propose to further investigate its use for assessing the pollution by co-
occurring non-nutrient pollutants, such as oils and heavy metals, which are entering
waterbodies simultaneously with nutrients during stormwater events.
Based on nutrient concentrations and stable isotope analysis, our data provide detailed
evidence that the lower Swan River estuary does not present a highly impacted urban





estuary. The nitrate stable isotope signature in the water suggested that the higher concentration of nitrate at two sites (MC, SCC) were due to a natural input of nitrate rather than human pollution. The stable spatial differences in mussel $\delta^{15}$N values over time that correlated to differences in nitrogen concentrations highlight the value of this organism as a bioindicator of spatial water quality assessment. Our data emphasizes that in systems with low pollution levels, the small differences in mussel stable isotope signatures reflect differences in site specific nutrient cycling caused by physicochemical conditions or biological factors rather than nitrogen pollution. This is important information for local management, but would have gone undetected at high pollution levels as the larger deviations of nitrogen stable isotope values would have made such small differences in mussel values invisible. We therefore advocate future studies in similarly (low) polluted systems that include stable isotope analysis of other food web end-members and nutrients of the groundwater, to develop an understanding of the baseline of spatial natural isotopic variability in urban aquatic systems.

In conclusion, this work shows the value of using stable isotope analysis as an integrative tool to establish an understanding on local processes and pollution levels in aquatic systems. In addition, we propose that it could help to define divisions in tidal estuaries based on natural characteristics and the human dimension that are meaningful for monitoring and management and for which reference conditions have to be identified (Ferreira et al., 2006).

**Acknowledgements**

This study was supported by a Research Development Award (2009) from the University of Western Australia to E.S. Reichwaldt and by an Australian Research Council Linkage Project (LP0776571) and the Water Corporation of Western Australia. The authors would like to thank S. C. Sinang, H. Song, and L. X. Coggins for help in the field and in the laboratory, L. X. Coggins for editing an earlier version of the manuscript, and C. Harrod for valuable help during the preparation of the manuscript. The permit for sampling mussels was obtained from the Department of Environment and Conservation, Western Australia (Licence no. SF007464). Discharge data were courtesy of the Department of Water, Western Australia.





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



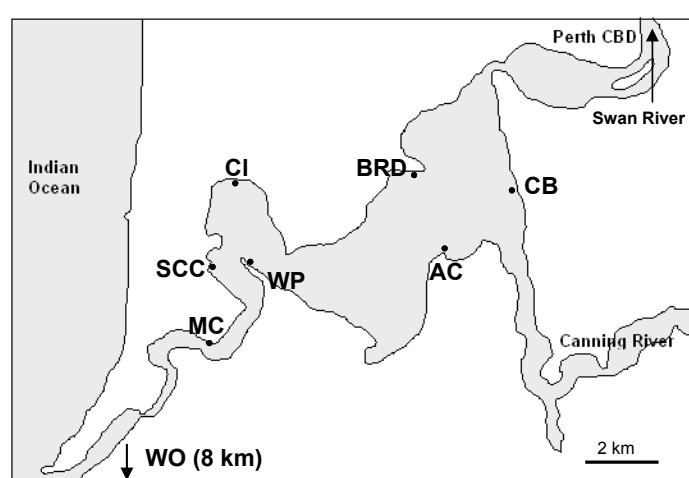

Figure 1. Map indicating the 7 sampling sites (jetties) within the Lower Swan River
estuary, Perth, Western Australia. AC = Applecross, BRD = Broadway, CB = Como
Beach, Cl = Claremont (Freshwater Bay), MC = Minim Cove, SCC = Swan River Canoe
Club, WP = Point Walter; the ocean reference site was located 8 km south of the estuary
mouth (WO = Woodman Jetty).





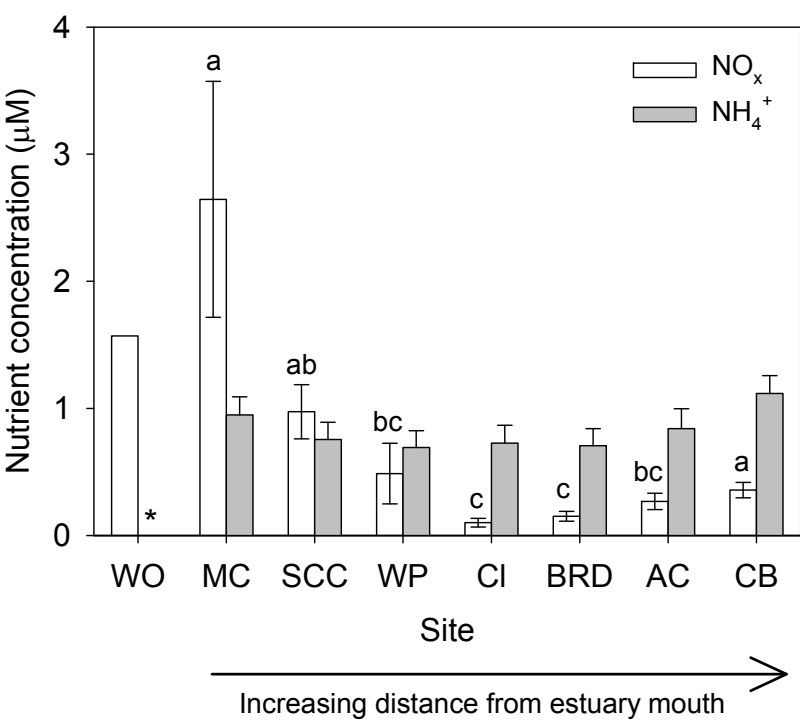

Figure 2. Mean concentration of $NO_x$ and $NH_4^+$ (μM) at each site. Letters indicate
differences between sites for $NO_x$ concentrations, with sites sharing the same letter being
not significantly different. Error bars represent one standard error (N = 17). Asterisk at
WO indicates that mean value of $NH_4$ was below the limit of quantification.





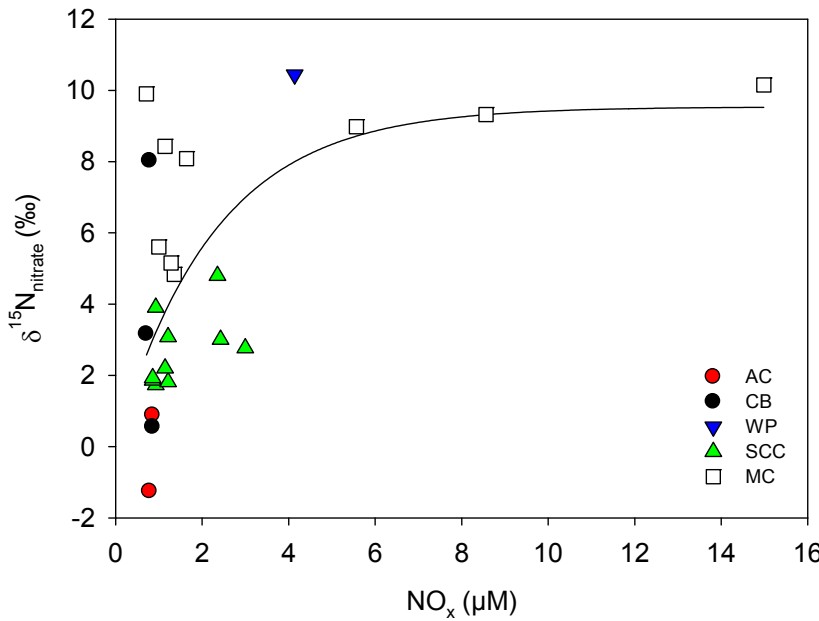

3  Figure 3. Relationship between nitrate $\delta^{15}N$ (‰) and the concentration of $NO_x$ (μM)

4  ($r^2 = 0.313$, $y = 9.54(1-e^{-0.44x})$).





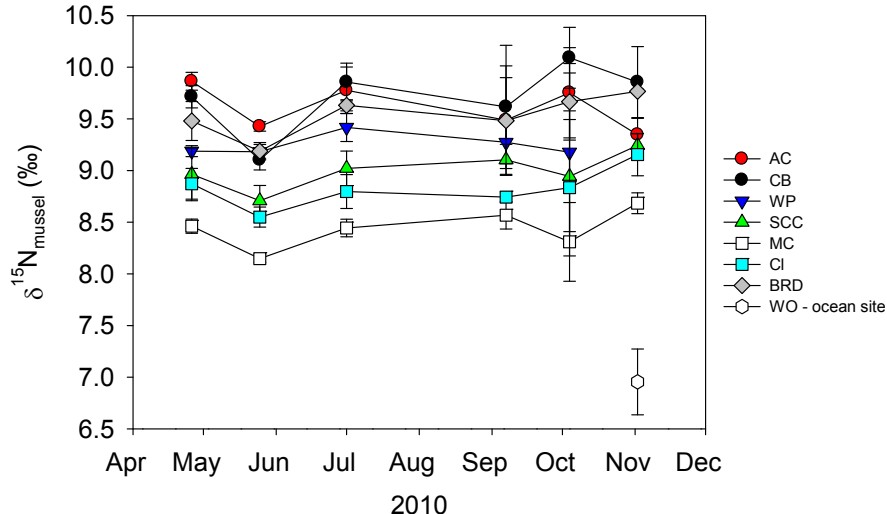

2    Figure 4. Mean $\delta^{15}$N mussel signature (‰) at each site over time. Error bars represent

3    standard deviations of N = 3 for April to July and WO, and N = 2 for September to

4    November 2010.



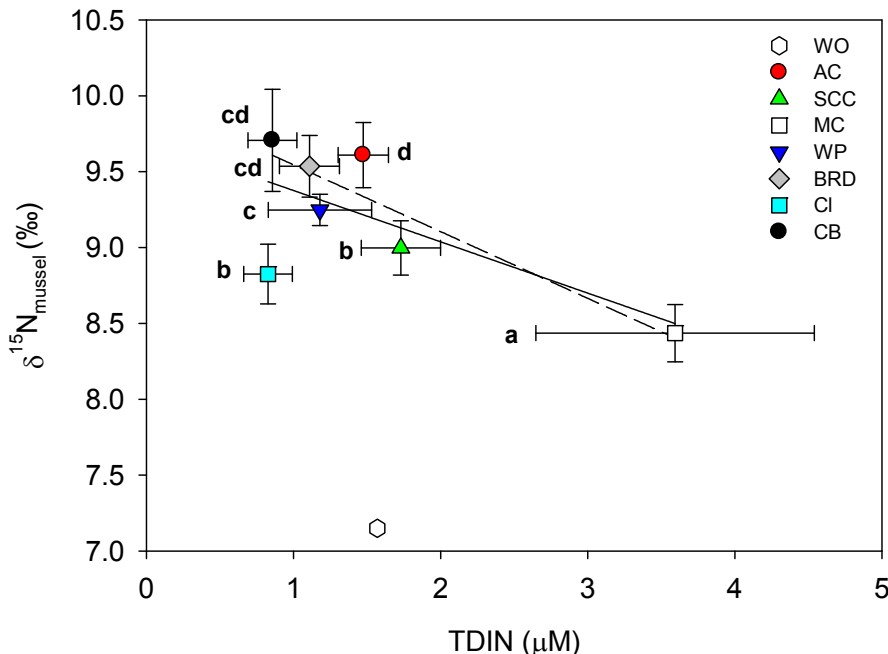

Figure 5. Relationship between mean mussel $\delta^{15}N$ (‰) and total dissolved inorganic
nitrogen (TDIN) ($\mu$M). Error bars represent standard deviation for mussels (N = 6 for all
sites except for WP where N = 5) and standard error of for TDIN (n = 17). The solid line
represents the relationship calculated for all sites ($r^2$=0.486, y=-0.338x+9.71), the broken
line when site Cl is omitted ($r^2$=0.838, y=-0.440x+9.98). Letters indicate differences in
$\delta^{15}N_{mussels}$ (ANOVA with Games Howell post hoc test), with sites sharing the same letter
being not significantly different.





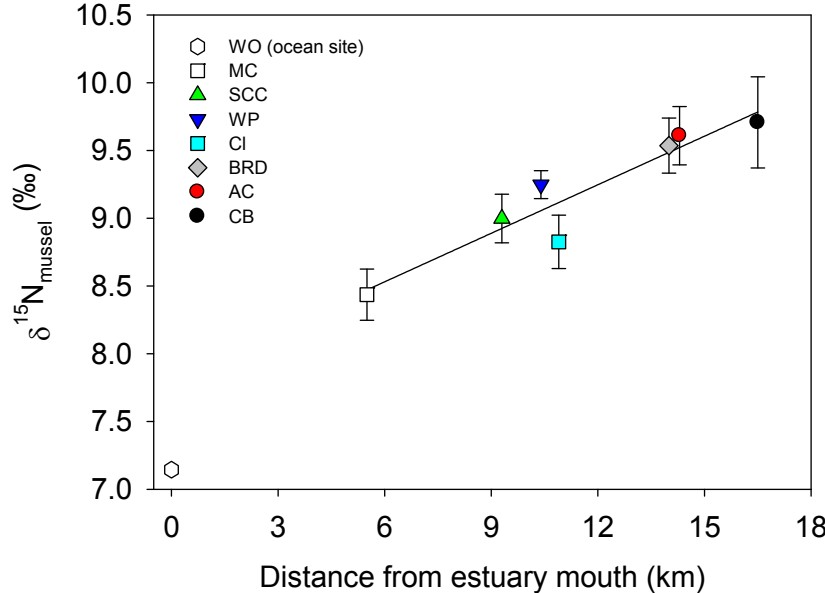

2    Figure 6. Relationship between mean $\delta^{15}$N of mussels (‰) and distance of sites from

3    estuary mouth ($r^2 = 0.563$, $y = 0.12x + 7.74$). Error bars represent standard deviation of

4    N = 6 for all sites except for WP where N = 5.





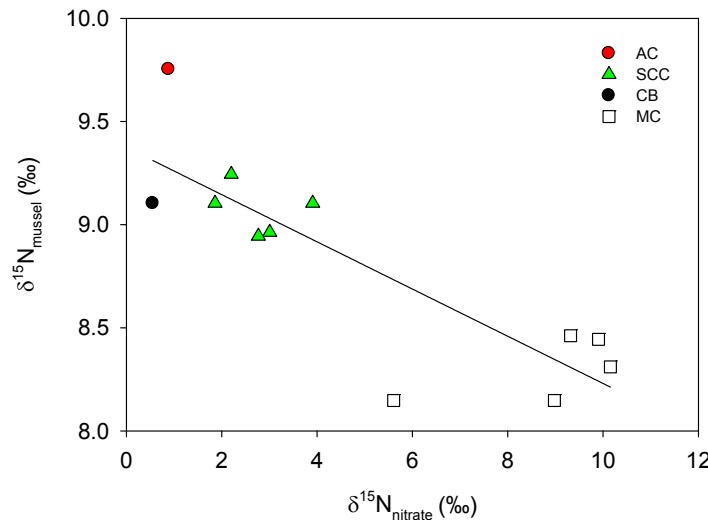

2    Figure 7. Relationship between nitrogen stable isotope signature of mussel and nitrate in

3    the water ($r^2 = 0.711$, y = -0.114x + 9.37).

