# Peer review of "Can mussels be used as sentinel organisms for"

_Hydrology and Earth System Sciences, 2015_

## Referee Comment (RC1) · Anonymous Referee #1 · 19 Feb 2016

Comments regarding paper Review of "Can Mussels be used as sentinel organisms for characterization of pollution in urban water systems?"

Overall

Well written and with an easy to follow set-up and results. I think the objectives could be improved and some of the introduction/discussion regarding water management approaches streamlined to move the reader more quickly to the meat of the paper.

I was a bit confused by exactly what was meant by pollution – nitrate or nitrogen generally. There is a strong focus on nitrate but the results don't point strongly to mussels reflecting nitrate concentration or 15N composition and therefore a larger focus on the

[Figure]

N-cycle may be needed to explain the results observed here.

I think with some slight reorganisation, streamlining and expansion of N-cycling within the estuary this manuscript will be much improved. More attention should be paid to POM and how/why or whether POM is decoupled from NO3 and how this relates to the 15N of the mussels. Your strongest figure is 15N-mussel vs the distance from estuary (some of others are strongly influenced by one site, MC) and this is not fully explained in the discussion.

There is a strong emphasis on site-specific characteristics influencing mussels however, aside from MC, the concentration of NO3 and NH4 were fairly low and not correlated across sites. So, NO3 and NH4 not likely to explain site-specific 15N-mussel variability. This should be addressed quickly in the discussion section.

With some tightening and expansion in some areas (my edits below) I think this paper constitutes a nice addition to the applicability of food webs and biological indicators of nutrient sources.

Page 1:

19 higher nitrogen stable isotope signature. Enriched in 15N is more accurate. Purely preference here, you did well 23 Can you omit the sentence beginning with "Our results showed a trend..." I think the sentence isn't necessary in the abstract. 26 What are natural values? Maybe state within range of observed values within estuaries of W. Australia 28 Delete " which allowed for the detection of spatial difference" 29 change to 'organisms'

Page 2:

First paragraph doesn't relate well to abstract or title. I would introduce this paper with the current state of affairs regarding nitrogen in urban water systems, then identify the problem – the quantification of a spatial and temporally varying regulated chemical species (nitrogen).

Second paragraph starts from the point of restoration and then proceeds to the problem – limited understanding of temporal and spatial variability of pollution (I would state nitrogen here, it is your focus)

I suggest deleting most of paragraph 1 and improving paragraph 2 to more concisely state your research problem, question etc. Get to the point of the paper very quickly.

Page 3:

Good. Paper is fully into the nitrogen sources, how to characterize variability etc.

11 – delete s from 'urbans'

Page 4:

18 Citation for this? Would be useful to reader to know how work in polluted water-bodies then relates to concentrations and isotopic composition presented in this paper – were polluted waterbodies exhibiting higher concentrations and heavier 15N values? Over a larger range? Etc

Make sure objectives use same phrasing. "Would lead to..." is good and used in 2 of 3. Keep it uniform to help the reader. (2) is more of a conclusion

26 change to 'nitrogen-rich'

27 '(2) distinct spatial difference in mussels...' This doesn't quite make sense. Do you mean to say that the number of mussels relates to the nitrate concentration? Or that the 15N composition of mussels reflect observed composition in nitrate.

29 'lead to increased anthropogenic signal'. Rephrase, you anticipate that you will observe elevated 15N due to elevated 15N inputs from nitrogen-rich waters, which follows your prediction (1).

8 Change to - prone to 'nutrient' pollution.

General – clarify that the 15N composition is reported in units relative to an international standard (air usually). I assume the standard is the same for both isotope facilities used in this paper. Report it and clarify that the 15N concentrations you report are relative to the standard and are not absolute concentrations (isotope scientists know this, others may not). Same for 18O. This should be done in the methods section at a minimum, often re-stated in data tables as part of the units of 15N

1 Change to "To determine the isotopic composition of nitrogen in particulate organic matter (POM), a source of food for mussels, 0.7 – 2.5 L. . .." Avoid using 'signature' unless you've determined that the isotopic composition of POM is unique, particularly if you're only using one isotope for characterization.

1b Your hypothesis #2 is that mussel 15N corresponds to nitrate 15N, no? But here you say that mussels feed on POM so the reader is confused by the nitrate 15N hypothesis. You should rectify this earlier in the introduction somehow. Either focus on POM or state how N cycling would link nitrate and POM 15N composition.

1c You state that mussel 15N and POM 15N are linked but you don't show in a figure. And the link between 15N POM and 15N NO3 is also not discussed in the results.

4 Change to "Harvested mussels were measured and dissected to obtain the foot tissue. . ."

6 Was the foot tissue homogenized before isotope analysis or was the entire sample of 3 combined foot tissue used in the mass-spectrometer? If the entire sample was used, state so, if the sample was fully homogenized with mortar/pestle state that. As it is it seems there were 3 distinct pieces of foot tissue were dried together.

4 long term average based on how many years? Citation?

7 The comparison is between discharge during the winter of 2010 and the winter of 1994 and the conclusion is that 2010 discharge was lower than usual. Is there a published mean discharge value you can compare to? Or is the discharge of '94 the only published value for comparison? To state discharge is lower than usual you should have an average or trend of some sort for comparison

10 Unusually high salinity? Is this relative to a published average salinity value for the estuary? Need citation or cleaner text. Either state the salinity was high throughout the area or high relative to a specific mean value (with citation if possible).

10b What are the units for salinity? I suggest adding the salinity recorded for the ocean water in the nearby area (or salinity of ocean water generally) for the reader to compare.

31 Change to "while nitrogen from $NH_4^+$ was greater at all other sites (Fig. 2)".

31 Can omit sentence starting with "This is supported by significant...". It doesn't add much value compared to previous sentence.

4 change to "The TN:TP ratio (weight) was between 0 and 6.5, with 84% of the ratios (by site? ) below 2.2". Move the rest of the paragraph to appropriate place in discussion OR condense to simple sentence that cites published thresholds for determining nitrogen limitation (7.2 or 2.2).

18 "Analysis of stable isotope composition of $NO_3$..." Change 'signature' throughout unless you're really talking about the uniqueness of a component's isotopic composition.

19 restate minimum concentration requirements

1 Clarify sentence findings – I understood that POM 15N and mussel 15N collected at each site had a significant, positive relationship to one another. By fractionation effect of 0.6 do you mean that mussel 15N composition was on average 0.6 greater than POM 15N composition at same site? Clarify this for the reader, particularly if you're not including a figure.

5 Move this sentence second in the paragraph. Move second sentence to the first sentence position.

7 'smaller than range seen in 15N nitrate' (... to ...) restate range of nitrate 15N to make it easier for the reader to compare the relative ranges of each.

8 use lower case , not $\Delta$. It would be better to rephrase the sentence so you are not starting with a greek letter.

9 "no temporal trend" sentence starts with a non-trend and ends with a significant (?) trend between 15N and distance to estuarine mouth, connect the two clauses with a 'though'.

Figures 3, 5 and 7 all strongly influenced by MC site.

Figure 5. You show scenarios with and without CI or MC sites, was WO site included in regressions?

31 avoid using 'site-specific' twice in same sentence. Restructure.

It would be easier for the reader if the discussion followed directly from the 3 objectives stated in the introduction – nitrogen and 15N conc upstream; 15N mussel by site and nitrate conc; distance from mouth = anth signal.

24 What do you mean by this sentence. Expand more. How does the fraction of NOx in the DIN pool explain site-specific variation in 15N? It's stated here but the reader doesn't understand how simply from the sentence.

First two sentences are redundant, simplify and merge. Sentence 1 is cumbersome with overuse of "15N values".

Trend between mussel 15N and nitrate 15N strongly driven by site MC. As is relationship with TDIN. Without MC site, there is little to no trend. You should address this head-on in your discussion section.

21 Relationship can't be 'higher'. The r2 value can be higher, the relationship can be stronger etc. Though the slope of the line doesn't change much with removal of CI site, the fit improves. I mention earlier but you should also clarify if you keep the WO site in the regression.

21b Good explanation of N cycling dynamics at this site. Could you include something similar for the MC site, even if it's conjectural it would be useful given how different the site was relative to the others.

POM and mussel 15N are linked but nitrate 15N negatively linked to mussel 15N (driven by MC site). Could it be that POM sources are not within-estuary? If you're estuary is N-limited then production should be low, could be that POM is all sourced outside (upstream I imagine) and within-estuary nitrate 15N and nitrate concentrations aren't important to POM production. This could explain uncoupled 15N between POM and NO3. Do you have evidence of this? This would still be in line with the overall story here, reinforcing need for site-specific information and management approaches.

18 but your nitrogen sources of nitrate and ammonium were not different between sites (except MC) so this seems unlikely to explain differences in mussel 15N, no? More likely differences in POM 15N drove differences in mussel 15N and is reflected in relationship between mussel 15N and distance from mouth.

It seems like there are other n cycling effects that are occurring here and could help

to explain the negative (or lack of) correlation between 15N-NO3 and 15N-mussel (or TDIN and 15N-mussel. Your MC site may be influencing interpretation too much. If you had to interpret these data without the MC site, how would you do so? Does it change your overall conclusions? I would like to see more results regarding the POM and its connection to N-cycling. You have a fairly strong trend between 15N-mussel and distance from estuary mouth. What is driving this?

4 correlated to nitrogen concentrations. But these were all negative correlations, no? TDIN is shown, not POM or NOx.

15N-mussel negatively correlated to 15N-NO3. 15N-NO3 positively (though MC weighs heavily) related to NOx concentration. High NO3 reflected in 15N-NO3 does not appear in 15N-mussels as sites with high 15N-NO3 and NO3 have low 15N-mussels, no? So mussels don't appear to be good indicators of NO3 sources as they don't reflect 15N-NO3, no?

In the discussion and conclusion sections you refer to mussels reflecting nitrate pollution but the link is weak, dependent on MC, and negative. Explain how these connections interact or simplify your message in the discussion and conclusion. The emphasis appears to be on nitrate but the linkages between nitrate and mussel tissue are unclear.

---

## Referee Comment (RC2) · Anonymous Referee #2 · 19 Feb 2016

General comments: [Content] This manuscript sets as an overarching framework the increasing pollution of water bodies located within or in the vicinity of urbanised areas. The authors have carried out a kind of proof-of-concept analysis where they investigate the potential for mussels to serve as an archive of nitrogen stable isotope signatures – as a proxy for spatial and temporal variability of nutrient pollution in the urban and tidally influenced estuary of the Swan River in Western Australia. Their results show that d15N signatures in mussels do not change significantly over time – thereby suggesting that they are very much site specific. The authors conclude from their findings that mussels have the potential for becoming a robust tool for detecting and characterising aquatic pollution in urban environments.

[Structure] The article is well written and structured. When going first through the manuscript, I had the impression that the introduction was pretty long (it is almost $\frac{1}{4}$ of the text). Having said that, there is a lot of useful information and references included. One option could be to shorten a bit the introduction, or to introduce a few sub-headings in order to make it an easier read: basically it is about (1) increasing human impact on aquatic ecosystems, (2) the need for a better understanding of the spatial and temporal variability of pollution levels with a view to better manage these often irreversibly impacted systems, (3) the focus on nutrient pollution, (4) the use of stable isotopes (especially of N) for investigating anthropogenic nutrient pollution, and (5) the introduction of mussels as a sentinel organism in that specific context. The remaining parts of the manuscript are well structured – the number of figures is appropriate.

Specific comments: 1-Introduction [pages 2-4]: When reading the introduction, and more specifically the paragraphs to the end where mussels are introduce as sentinel organisms, I was surprised (unless I am mistaken) not to learn about what species have eventually been used for this study. I think this is a very important aspect that the authors have not taken into consideration for their manuscript. In an area where they expect living organisms to be a living archive of the local average environmental conditions it is essential to know a minimum about the metabolism of that organism. Especially in a journal that has a large community of readers from hydrological sciences, we cannot necessarily expect them to know much about this topic. Moreover, since this is a kind of proof-of-concept study, the authors should carefully describe the organisms, growth rates, sensitivity to changing environmental conditions etc. These aspects are likely to be crucial when it comes to eventually understand and discuss the isotopic signatures of N in the mussel's foot tissue. As mentioned further down in this assessment, there is existing literature in this respect and it would certainly be of value to take this into consideration in a revised version of the manuscript.

2-Material and methods [page 5 study sites & 6 sampling and analyses]: When read-
ing the changing conditions in the Swan River estuary, one could expect differences between mussel species that are exposed to these fluctuations in salinity (between high tide and low tide). Is there only one mussel species in the studied area? If not (what is very likely), what are the other species that are present – what species has the sampling protocol been targeting – was it a mix of species – how sure can we be that different sensitivities to changing environmental conditions (including pollution) can lead to differences in metabolic activity? A few examples of literature along these lines can be: - Atkinson et al., 2010. Stable isotopic signatures, tissue stoichiometry, and nutrient cycling (C and N) of native and invasive freshwater bivalves. Journal of the North American Benthological Society 29(2):496-505. - Gustafson et al., 2007. Temporal and spatial variability in stable isotope compositions of a freshwater mussel: implications for biomonitoring and ecological studies. Oecologia 152: 140-150. - Hawkins, A.J.S., Bayne, B.L., 1985. Seasonal variation in the relative utilization of carbon and nitrogen by the mussel Mytilus edulis : budgets, conversion efficiencies and maintenance requirements. Mar. Ecol. Prog. Ser. 25(2): 181-188

3-Results [page 8 physicochemical parameters]: given that the study was carried out during rather dry conditions, the prevailing environmental parameters measured in the investigated area have also been rather unusual as stated in the manuscript. Here again, it would be interesting to see how the mussels populations have responded to that (if at all) – is there any information available on that? On page 8, line 10 units should be added to salinity. On page 10 the delta symbol should be homogenised.

4-Discussion [page 13]: In lines 6 to 8 I would be careful when stating that stable isotope signatures in mussels of tidally influenced estuaries are less impacted by seasonal changes in watershed input and chemistry compared to large rivers. This statement make sense considering the results of this study, but given the particularly dry conditions that prevailed during this investigation and the proof-of-concept character of this study, there need most probably to be more investigations before a strong statement in this sense.

5-Conclusion [page 13]: A similar comment as for the point above can be made for the 1st paragraph of the conclusion. Again the lack of information on the studies species, their metabolism, etc comes into play here. Of interest could also be to see if there are differences in signatures between species. In the conclusion it is stated that the future studies should contribute in similar (low) polluted systems to better understand the baseline of spatial natural isotopic variability in urban aquatic systems. I was wondering if this is not somehow contradictory with what is announced in the title – are mussels then really used in the sense of sentinels of pollution or rather as indicators of the baseline of 'spatial natural isotopic variability in urban aquatic systems'. Again here I am possibly confused by the fact that no information is given on how sensitive those organisms are eventually to pollution. As a last comment, one could also say that nutrient pollution is not really an urban problem or at least the origin of it can most of the time be found further upstream in agricultural parts of the catchments. In urban environments, one could also be targeting other sources of pollution, such as heavy metals, xenobiotics, etc.

Concluding remarks: This manuscript is certainly a very interesting contribution for the readers of this journal and I enjoyed very much reading it. It is an interesting case study – or more specifically a proof-of-concept study – introducing mussels as a sentinel organism for investigating nutrient pollution in an urban aquatic environment. Since existing literature on similar applications/studies is not much referred to in the manuscript, the innovative character of this study might however be slightly overrated.

Given the assessment provided above, this manuscript should undergo – prior to publication – what should be considered moderate to major revisions.

---

## Author Comment (AC1) · 8 Mar 2016

Dear Referee #1,

Thank you very much for your very insightful and positive comments on our manuscript "Can mussels be used as sentinel organisms for characterisation of pollution in urban water systems?" by E. S. Reichwaldt and A. Ghadouani (hess-2015-523). They provide very important feedback to improve this manuscript.

Below is a proposed revision report on how we would like to address your comments.

We will first reply to your overall comments, before we will address the specific comment in more detail.

**1) Overall comments**

| No. | Comments | Response |
|---|---|---|
| 1 | I think the objectives could be improved | We agree and have restated the objectives using your suggestion from your comment #14. The objectives will now read as follows: "Specifically, we anticipated that (1) a higher input of nitrogen-rich waters upstream would lead to a higher isotopic signatures of nitrate, (2) spatial differences in the level of nitrates in the water would lead to spatial differences in mussel isotopic signature, and (3) the increased distance from the mouth would lead to an elevated $^{15}$N values in mussels due to elevated $^{15}$N inputs from nitrogen-rich waters upstream. " |
| 2 | Some of the introduction/discussion regarding water management approaches streamlined to move the reader more quickly to the meat of the paper. | This comment is a short version of your comment #11 and we will reply in more detail later. In brief, we will rework the introduction along your suggestions, but would like to point out that a second reviewer found this introduction very useful. As such, we will not delete the complete first paragraph of the introduction, but at this stage consider deleting some sentences (e.g., Page 2 Lines 9 – 13 and the first sentence of the second paragraph) and exchanging the words "nutrient" and "pollutant" with "nitrogen" to make it more specific in the third paragraph. For more details, please see our reply to your comment #11. |

| 3 | I was a bit confused by exactly what was meant by pollution – nitrate or nitrogen generally. There is a strong focus on nitrate but the results don't point strongly to mussels reflecting nitrate concentration or 15N composition and therefore a larger focus on the N-cycle may be needed to explain the results observed here. | We agree that this needs clarification. The analysis of the nitrogen signature in general has proven to be a powerful tool as an indicator of anthropogenic contamination. Our study looks specifically at the stable isotope signature of nitrate. Then, to additionally test if mussels can be used as bioindicators of nitrogen pollution, we broaden our objectives and look at nitrogen. |
|---|---|---|
| | | We will make this clearer throughout the manuscript by exchanging the word "nutrients" with "nitrogen" where appropriate by restating the objectives: "The main aim of this study was to identify the variability of nitrogen concentration in an urban estuary over time and space and to ascertain the suitability of the isotopic signature ($\delta^{15}N$) of mussel tissue as an indicator of nitrogen pollution in urban water systems. Specifically, we anticipated that (1) a higher input of nitrogen-rich waters upstream would lead to a higher isotopic signatures of nitrate, (2) spatial differences in the level of nitrates in the water would lead to spatial differences in mussel isotopic signature, and (3) the increased distance from the mouth would lead to an elevated $^{15}N$ values in mussels due to elevated $^{15}N$ inputs from nitrogen-rich waters upstream. |
| 4 | More attention should be paid to POM and how/why or whether POM is decoupled from NO3 and how this relates to the 15N of the mussels. Your strongest figure is 15N-mussel vs the distance from estuary (some of others are strongly influenced by one site, MC) and this is not fully explained in the discussion. | We agree and we will include a paragraph on page 12 Line 21as follows: "An alternative explanation would be that POM could originate upstream where nitrate might have had higher $\delta^{15}N$ values (not quantified in this study). Upon entering the estuary, POM mixes with estuarine POM, uncoupling the within-estuary $\delta^{15}N$ nitrate and POM $\delta^{15}N$ values. This could also explain the strong relationship between $\delta^{15}N$ in mussels and the distance from the estuary mouth found in our study. Such a strong relationship can be expected in estuaries with low pollution levels due to the aforementioned mixing, while little spatial variability in $\delta^{15}N$ values of primary consumers can be expected in heavily polluted estuaries due to the dominance of upstream POM, as was shown by Oczkowski et al. (2008)." |
| | | Please also see our reply to your comment #48. |

| 5 | There is a strong emphasis on site-specific characteristics influencing mussels however, aside from MC, the concentration of NO3 and NH4 were fairly low and not correlated across sites. So, NO3 and NH4 not likely to explain site-specific 15N-mussel variability. This should be addressed quickly in the discussion section. | We agree with the reviewer. Because NO3, NH4, or TDIN concentrations are very similar for many of the sites they cannot completely explain $\delta^{15}N$ variability in mussels. We will include this in

i) the discussion (Page 12 Line 31):" Site MC was closest to the ocean, was one of the deepest sites and had a higher TDIN concentration compared to all other sites, which were not different with regard to TDIN concentration between themselves. This emphasises that the differences in mussel $\delta^{15}N$ between sites might be due to site-specific nutrient cycling processes in our estuary and might not reflect nitrogen pollution itself."

ii) the conclusion (Page 14 Line 5):" The negative trends between mussel $\delta^{15}N$ values and nitrate concentration or nitrate $\delta^{15}N$ values emphasize that mussels might not be good indicators for $NO_3$ sources in systems with low pollution levels. Instead, the small differences in mussel stable isotope signatures might reflect differences in site specific nutrient cycling caused by physicochemical conditions or biological factors rather than nitrogen pollution."

Please also see our reply to your comment #50. |

**2) Specific comments**

| 6 | Page 1:
19 higher nitrogen stable isotope signature. Enriched in 15N is more accurate. Purely preference here, you did well | We agree that both phrases can be used. We prefer to keep "higher nitrogen stable isotope signature" in the abstract to make it easier for readers who are not entirely familiar with stable isotope jargon. |
| 7 | Page 1:
23 Can you omit the sentence beginning with "Our results showed a trend. . ." I think the sentence isn't necessary in the abstract. | We agree and will omit this sentence in the abstract. |

| 8 | Page 1:
26 What are natural values? Maybe state within range of observed values within estuaries of W. Australia | We agree that this was ambiguous. We will clarify it by rewriting this sentence to :" … nitrogen stable isotope values of nitrate throughout the estuary were well within natural values of uncontaminated groundwater or organic nitrate from soils, indicating groundwater inflow rather than pollution by human activity was responsible for differences between sites."

This will and has been described in more detail with references in the discussion on page 11 Lines 15-23.

Unfortunately no values for Western Australia are available, but we will additionally provide a citation for this statement in the conclusion: Page 14 Line 2: "…were due to a natural input of nitrate uncontaminated groundwater (Xue et al., 2009) rather than human pollution.". |
|---|---|---|
| 9 | Page 1:
28 Delete " which allowed for the detection of spatial difference" | We agree with this comment and will delete this part of the sentence. |
| 10 | Page 1:
29 change to 'organisms' | We agree with this comment and will correct this. |
| 11 | Page 2:
First paragraph doesn't relate well to abstract or title. I would introduce this paper with the current state of affairs regarding nitrogen in urban water systems, then identify the problem – the quantification of a spatial and temporally varying regulated chemical species (nitrogen).
Page 2:
Second paragraph starts from the point of restoration and then proceeds to the problem – limited understanding of temporal and spatial variability of pollution (I would state nitrogen here, it is your focus) I suggest deleting most of paragraph 1 and improving paragraph 2 to more concisely state your research problem, question etc. Get to the point of the paper very quickly. | We understand your concern and will aim to strike a good balance between "getting to the point quickly" and "giving a broad picture of state of pollution management", which we think is appropriate for this journal that has such a broad range of readers. We therefore consider keeping the first two paragraphs as the general introduction but in a shorter version. We think that by applying the following, we will strike a nice balance between your suggestions and the comment from the second reviewer, who thought our introduction was very valuable:

i) delete part of the first paragraph (Page 2 Line 9 – 13)

ii) delete the first sentence of the second paragraph to make sure that we get to the point more quickly

iii) in the third paragraph, we will consider being more specific by exchanging the words "nutrient" and "pollutant" with "nitrogen". |

| 12 | Page 3:
11 – delete s from 'urbans' | We agree with this comment and will correct this. |
|---|---|---|
| 13 | Page 4:
18 Citation for this?

Would be useful to reader to know how work in polluted waterbodies then relates to concentrations and isotopic composition presented in this paper – were polluted waterbodies exhibiting higher concentrations and heavier 15N values? Over a larger range? Etc | The citations are currently included after the next sentence. We will bring them forward and combine the two sentences. It will then read as: "Earlier studies in polluted freshwater and marine systems found positive relationships between the concentration of nitrogen and the isotopic signature of nitrogen in mussels, and between the isotopic signature of nitrate-N and that of mussels (Cabana and Rasmussen, 1996; McClelland et al., 1997; Costanzo et al., 2001; Anderson and Cabana, 2005; Gustafson et al., 2007; Wen et al., 2010), suggesting that bivalves are suitable indicators of changes in nutrient pollution load to waterbodies."

We will also add, as suggested, a brief comparison between nitrogen concentration / $\delta^{15}N$ values found in our study and the values reported in the cited previous studies. We will add this on page 12 Line 3. |
| 14 | Page 4:
Make sure objectives use same phrasing. "Would lead to. . ." is good and used in 2 of 3. Keep it uniform to help the reader. (2) is more of a conclusion | We agree with his comment and will rephrase objective (2) as follows: "(2) spatial differences in the level of nitrates in the water would lead to spatial differences in mussel isotopic signature" |
| 15 | Page 4:
26 change to 'nitrogen-rich' | We agree with this comment and will correct this. |
| 16 | Page 4:
27 '(2) distinct spatial difference in mussels. . .' This doesn't quite make sense. Do you mean to say that the number of mussels relates to the nitrate concentration? Or that the 15N composition of mussels reflect observed composition in nitrate. | We agree that this was expressed ambiguously. We will restate objective 2 as suggested above (#14) to "(2) spatial differences in the level of nitrates in the water would lead to spatial differences in mussel isotopic signature" to make this explicit. |
| 17 | Page 4:
29 'lead to increased anthropogenic signal'. Rephrase, you anticipate that you will observe elevated 15N due to elevated 15N inputs from nitrogen-rich waters, which follows your prediction (1). | We agree and will rephrase objective 3 as follows: "(3) the increased distance from the estuary mouth would lead to elevated [15]N values in mussels due to elevated [15]N inputs from nitrogen-rich waters upstream." |

| 18 | Page 5
8 Change to - prone to 'nutrient' pollution | We agree and will add the word "nutrient" into the sentence. |
|---|---|---|
| 19 | Page 6
General – clarify that the 15N composition is reported in units relative to an international standard (air usually). I assume the standard is the same for both isotope facilities used in this paper. Report it and clarify that the 15N concentrations you report are relative to the standard and are not absolute concentrations (isotope scientists know this, others may not). Same for 18O. This should be done in the methods section at a minimum, often re-stated in data tables as part of the units of 15N | We agree with this comment and can confirm that both institutions used the same standards ($\delta^{15}$N: air; $\delta^{18}$O: Vienna Standard Mean Ocean Water, VSMOW) and that all values are reported in per mil (‰) with respect to the international standards. We will include the following sentences in the method section:

- mussels and POM: "All values are reported in per mill (‰) with respect to the international standard (air)."

- nitrate: "All values are reported in per mill (‰) with respect to the international standards ($\delta^{15}$N: air; $\delta^{18}$O: Vienna Standard Mean Ocean Water, VSMOW)." |
| 20 | Page 7
1 Change to "To determine the isotopic composition of nitrogen in particulate organic matter (POM), a source of food for mussels, 0.7 – 2.5 L. . .." Avoid using 'signature' unless you've determined that the isotopic composition of POM is unique, particularly if you're only using one isotope for characterization. | We agree and will change the sentence as suggested to "To determine the isotopic composition of nitrogen in particulate organic matter (POM), the food source for mussels that presents the direct link between nitrate and the mussels, 0.7 – 2.5 L…". |
| 21 | Page 7
1b Your hypothesis #2 is that mussel 15N corresponds to nitrate 15N, no? But here you say that mussels feed on POM so the reader is confused by the nitrate 15N hypothesis. You should rectify this earlier in the introduction somehow. Either focus on POM or state how N cycling would link nitrate and POM 15N composition. | We agree that we have made the link between [15]N of nitrate, POM and mussels not clear enough. We will therefore explain this better in the introduction by adding the following sentences:

Page 4: Line 4: "This signal is then passed on to higher trophic levels up the food chain (Cabana and Rasmussen, 1994; Carvalho et al., 2015): Elevated $\delta^{15}$N signals in nitrate have been shown to lead to elevated $\delta^{15}$N signals in organisms that directly take up nitrate from the water, such as phytoplankton and microbes (Harrington et al., 1998). These organisms form an important part of particulate organic matter (POM), which serves as food for filter feeders (e.g., mussels). Mussels that ingest POM with elevated $\delta^{15}$N signal will then also show a higher $\delta^{15}$N signal." |

| | | Page 7 Line 1: "To determine the isotopic composition of nitrogen in particulate organic matter (POM), the food source for mussels that presents the direct link between nitrate and the mussels,…" |
|---|---|---|
| 22 | Page 7
1c You state that mussel 15N and POM 15N are linked but you don't show in a figure. And the link between 15N POM and 15N NO3 is also not discussed in the results. | We believe that you refer to Page 9.

We agree and will include the figure showing the significant, positive relationship between mussel and POM $\delta^{15}$N and we think it will be helpful for the reader, because in the improved manuscript version we will discuss more results regarding POM and its relationship to mussel nitrogen stable isotope values. (see also your comment #48)

We will also include a sentence on the relationship between $\delta^{15}$N of POM and nitrate. This will read as follows: "The relationship between $\delta^{15}$N of POM and nitrate was not significant; however this calculation was based on only five data points where simultaneous measurements of the two $\delta^{15}$N values were available, making this result arguable." |
| 23 | Page 7
4 Change to "Harvested mussels were measured and dissected to obtain the foot tissue. . ." | We agree and will change it as suggested to: "Harvested mussels were measured and dissected to obtain the foot tissue for stable isotope analysis." |
| 24 | Page 7
6 Was the foot tissue homogenized before isotope analysis or was the entire sample of 3 combined foot tissue used in the mass-spectrometer? If the entire sample was used, state so, if the sample was fully homogenized with mortar/pestle state that. As it is it seems there were 3 distinct pieces of foot tissue were dried together. | We agree that the description was unclear. We will therefore change it to:

"The feet of three individuals per site were combined, dried at 60°C for at least 24 h, fully homogenized with mortar/pestle, and stored in a desiccator until a subsample was analysed for mussel $\delta^{15}$N and C:N ratio." |
| 25 | Page 8:
4 long term average based on how many years? Citation? | We agree and will add the years the average was based on and the citation. The sentence will now read "Rainfall was below average in 2010 with 421 mm for the entire sampling period, while the average for this period for the previous 17 years was 690 mm (1993-2009; Bureau of Meteorology, 2016)." |

| 26 | Page 8:
7 The comparison is between discharge during the winter of 2010 and the winter of 1994 and the conclusion is that 2010 discharge was lower than usual. Is there a published mean discharge value you can compare to? Or is the discharge of '94 the only published value for comparison? To state discharge is lower than usual you should have an average or trend of some sort for comparison. | We agree with this comment and will report the average discharge for 1993-2009 and the minimum and maximum values within this period. These values are taken from the Department of Water data base, which we will cite. It will now read as follows: "This resulted in a lower than usual discharge from the tributaries into the estuary with a mean discharge from the Swan River of $1.2 \times 10^5 \text{ m}^3 \text{ d}^{-1}$ in 2010 compared to an average discharge of $8.4 \times 10^6 \text{ m}^3 \text{ d}^{-1}$ for the period of 1993-2009 for the same season (min. – max: $1.99 \times 10^6 \text{ m}^3 \text{ d}^{-1}$ (2002) – $2.21 \times 10^7 \text{ m}^3 \text{ d}^{-1}$ (1996) (Department of Water, 2016)." |
|---|---|---|
| 27 | Page 8:
10 Unusually high salinity? Is this relative to a published average salinity value for the estuary? Need citation or cleaner text. Either state the salinity was high throughout the area or high relative to a specific mean value (with citation if possible).

10b What are the units for salinity? I suggest adding the salinity recorded for the ocean water in the nearby area (or salinity of ocean water generally) for the reader to compare. | We agree with this comment and will add a citation from a previous study, which reports on salinity in this estuary. We will also add that seawater has a salinity of 35. The section will now read: "This might have contributed to higher salinities throughout the entire estuary during this study than previously reported (Stephens and Imberger 1997) and no relationship between salinity and distance to the estuary mouth was detected. During high tide, the salinity at all sites was between 24.2 and 32.4 and there was no difference in salinity between sites which can be considered brackish to saline (salinity of seawater is 35)."

We believe that this information together with the description of the Swan River Estuary (2.1 Study site) will now be sufficient to understand the dynamics of salinity in this estuary.

We would like to note that salinity does not have a unit as it is a ratio of the conductivity of a seawater sample and a standard potassium chloride solution (see UNESCO (1985): The international system of units (SI) in oceanography, UNESCO Technical Papers No. 45, IAPSO Pub. Sci. No. 32, Paris, France.) We will therefore not include a unit. |
| 28 | Page 8:
31 Change to "while nitrogen from NH4+ was greater at all other sites (Fig. 2)". | We agree and will change it as suggested. |

| 29 | Page 8:
31 Can omit sentence starting with "This is supported by significant. . .". It doesn't add much value compared to previous sentence. | We agree and this section will now read: "On average, $NO_x$ was the dominant N source at MC, SCC and WO, while nitrogen from $NH_4^+$ was greater at all other sites (Fig. 2) (Kruskal Wallis one-way ANOVA, H = 59.0, df = 6). |
|---|---|---|
| 30 | Page 9:
4 change to "The TN:TP ratio (weight) was between 0 and 6.5, with 84% of the ratios (by site? ) below 2.2". Move the rest of the paragraph to appropriate place in discussion OR condense to simple sentence that cites published thresholds for determining nitrogen limitation (7.2 or 2.2). | We agree and we will change this to: "The TN:TP ratio (weight) was between 0 and 6.5 with 84% of the samples in our study being below 2.2, indicating a high possibility of nitrogen limitation in this system (Redfield 1958; Geider and La Roche, 2002). |
| 31 | Page 9:
18 "Analysis of stable isotope composition of NO3. . ." Change 'signature' throughout unless you're really talking about the uniqueness of a component's isotopic composition. | We agree and will change this as suggested here and elsewhere throughout the manuscript (e.g., Page 12 Line 30). |
| 32 | Page 9:
19 restate minimum concentration requirements | We agree and will include the minimum concentration requirement. It will now read as: "Analysis of the stable isotope composition of $NO_3$ was limited to a total of 25 samples that fulfilled nutrient concentration requirements for the analysis (0.71 µM $NO_3$-N)." |
| 33 | Page 10:
1 Clarify sentence findings – I understood that POM 15N and mussel 15N collected at each site had a significant, positive relationship to one another. By fractionation effect of 0.6 do you mean that mussel 15N composition was on average 0.6 greater than POM 15N composition at same site? Clarify this for the reader, particularly if you're not including a figure. | We agree that this was unclear. We will delete the part about the fractionation, as it is not relevant for the message of the paper. |
| 34 | Page 10:
5 Move this sentence second in the paragraph. Move second sentence to the first sentence position. | We agree and will change the position of these two sentences. |

| | | |
|---|---|---|
| 35 | Page 10:
7 'smaller than range seen in 15N nitrate' (. . . to . . .) restate range of nitrate 15N to make it easier for the reader to compare the relative ranges of each. . | We agree and have changed this to: "Values of $\delta^{15}N$ of mussels varied between 6.8 and 10.3 ‰ and the range was therefore smaller than the range seen in nitrate $\delta^{15}N$ (-1.3 and 10.4 ‰)." |
| 36 | Page 10:
8 use lower case , not _. It would be better to rephrase the sentence so you are not starting with a greek letter. | We agree and will restate the sentence. It will now read as: "Mussel $\delta^{15}N$ was significantly different between sites (one-way ANOVA; $\delta^{15}N$: $F_{6,98} = 42.53$) (Fig. 5) and increased significantly with distance from the estuary mouth ($r^2 = 0.563$, $y = 0.12x+7.74$, $F_{1,110} = 141.65$) (Fig. 6)." |
| 37 | Page 10:
9 "no temporal trend" sentence starts with a non-trend and ends with a significant (?) trend between 15N and distance to estuarine mouth, connect the two clauses with a 'though'. | We believe that this comment refers to the following sentences (page 10 Lines 7-10): "No temporal trend in mussel $\delta^{15}N$ was detected (Fig. 4). $\Delta^{15}N$ of mussels was significantly different between sites (one-way ANOVA; $\delta^{15}N$: $F_{6,98} = 42.53$) (Fig. 5) and mussel $\delta^{15}N$ increased with increasing distance from the estuary mouth (Fig. 6)."

We will change the second sentence as suggested in the previous comment (#36) and also add that the increase with distance from the estuary mouth was significant. As the two sentences are two separate analyses(the first is a temporal, the second a spatial analysis), we will not combine these sentences. It will now read as: "No significant relationship between mussel length and mussel $\delta^{15}N$ (linear regression; $F_{1,13} = 2.235$) was found. No temporal trend in mussel $\delta^{15}N$ was detected (Fig. 4). Mussel $\delta^{15}N$ was significantly different between sites (one-way ANOVA; $\delta^{15}N$: $F_{6,98} = 42.53$) (Fig. 5) and increased significantly with distance from the estuary mouth ($r^2 = 0.563$, $y = 0.12x+7.74$, $F_{1,110} = 141.65$) (Fig. 6)." |
| 38 | Page 10:
Figures 3, 5 and 7 all strongly influenced by MC site. | We agree and will discuss the strong influence of MC on relationships at various places throughout the manuscript. For instance,

Page 9 Line 11: "The concentrations of total dissolved inorganic nitrogen …. were higher towards the estuary mouth (Fig. 2), although these relationships were weak …. and were driven by site MC only."

Page 11 Line 13: "In the Swan River estuary, $NO_3$ was enriched and there was a positive relationship between nitrate $\delta^{15}N$ and the concentration of |

| | | NO$_x$ throughout the estuary, although this was strongly driven by site MC." |
|---|---|---|
| | | Page 12 Line 7: "Our study showed a positive relationship between food (POM) and mussel δ$^{15}$N, but a negative relationship between nitrate δ$^{15}$N and consumers (mussels), which was strongly affected by site MC." |
| | | Page 12 Line 21: "The relationship between mussel δ$^{15}$N and TDIN concentration within the estuary was much stronger when omitting site Cl and not significant when omitting site MC." |
| | | In addition, we will include a paragraph in which we will interpret the data without site MC (Page 12 Line 31):" Site MC was closest to the ocean, was one of the deepest sites and had a higher TDIN concentration compared to all other sites, which were not different with regard to TDIN concentration between them. This emphasises that the differences in mussel δ$^{15}$N between sites might be due to site-specific nutrient cycling processes in our estuary and might not reflect nitrogen pollution itself." |
| | | This will not change our overall conclusion that mussels can be used as indicators for site-specific differences in pollution or nutrient cycling, which is "…important information for local management, but would have gone undetected at high pollution levels as the larger deviations of nitrogen stable isotope values would have made such small differences in mussel values invisible." (page 14 Line 8) |
| 39 | Page 10:
Figure 5. You show scenarios with and without CI or MC sites, was WO site included in regressions? | We did not include the marine site (WO) in the regression. We agree that this has not been described clearly and we will now include the following sentence in the figure legend: "WO was not included in the regressions." There are two reasons why we did not include WO in the regressions: 1) the N-cycle is likely to be different in the estuary compared to the marine environment; 2) The purpose of this paper is to identify if mussels can be used as bioindicators within a system, which would be the estuary in our case. As such, including the marine site is not relevant but would rather confound trends and findings. The purpose of showing WO is purely to |

| | | provide a baseline for a marine environment. |
|---|---|---|
| 40 | Page 10:
31 avoid using 'site-specific' twice in same sentence. Restructure | We agree and will substitute the second "site-specific" by "spatial". |
| 41 | Page 11
It would be easier for the reader if the discussion followed directly from the 3 objectives stated in the introduction – nitrogen and 15N conc upstream; 15N mussel by site and nitrate conc; distance from mouth = anth signal. | We agree and by adopting changes from this and your other comments, the discussion of our data will be structured as follows:

1) nitrogen concentrations in the estuary (spatial; upstream/downstream)

2) discussion of nitrate $\delta^{15}$N values (site specific; processes that lead to differences between these values).

3) Mussel $\delta^{15}$N between sites and relationship between nutrient concentrations

4) Mussels and distance from estuary mouth

5) Mussel $\delta^{15}$N over time and suitability as indicators |
| 42 | Page 11:
24 What do you mean by this sentence. Expand more. How does the fraction of NOx in the DIN pool explain site-specific variation in 15N? It's stated here but the reader doesn't understand how simply from the sentence | We agree that this paragraph was unclear and needed expanding. We will restate it as follows: "The fraction of NOx of the TDIN pool (%) was significantly different between sites (data not shown; y = 0.15x-6.9, $r^2$ = 0.215, $F_{1,23}$ = 6.30, P < 0.05), with site MC having a higher mean fraction (mean = 62.5%) compared to all other sites, except for SCC. An earlier study by Sugimoto et al. (2009) also found a positive relationship between nitrate $\delta^{15}$N values and the nitrate fraction in TDIN which they explained by *in situ* isotopic effects during nitrification. However whether higher $\delta^{15}$N values of nitrate at MC are related to site specific nitrification rates in our estuary needs further investigation, because the $\delta^{18}$O and $\delta^{15}$N values of nitrate are rather representative of atmospheric $NO_3$ deposition values (Durka et al., 1994; Fang et al., 2011) and nitrification is likely to play a minor role at ammonium concentrations <5 $\mu$M (Day et al., 1989) that prevail in the Swan River estuary. |
| 43 | Page 12
First two sentences are redundant, simplify and merge. | We agree and will restated these two sentence as following: "Earlier studies indicated that nitrogen $\delta^{15}$N values are reflected in higher trophic levels in a |

| | | |
|---|---|---|
| | Sentence 1 is cumbersome with overuse of "15N values". Trend between mussel 15N and nitrate 15N strongly driven by site MC. As is relationship with TDIN. Without MC site, there is little to no trend. You should address this head-on in your discussion section. | predictable way with a positive relationship between $\delta^{15}N$ of nitrate, primary producer and primary consumer (e.g., mussels) (Cabana et al., 1994; Cabana and Rasmussen, 1996; Harrington et al., 1998; Oczkowski et al., 2008; Carvalho et al., 2015)."

We will mention the fact that the trends are strongly driven by site MC in the following places (please also see our reply to your comment #38):

Page 9 Line 11: "The concentrations of total dissolved inorganic nitrogen …. were higher towards the estuary mouth (Fig. 2), although these relationships were weak …. and were driven by site MC only."

Page 11 Line 13: "In the Swan River estuary, $NO_3$ was enriched and there was a positive relationship between nitrate $\delta^{15}N$ and the concentration of $NO_x$ throughout the estuary, although this was strongly driven by site MC."

Page 12 Line 7: "Our study showed a positive relationship between food (POM) and mussel $\delta^{15}N$, but a negative relationship between nitrate $\delta^{15}N$ and consumers (mussels), which was strongly affected by site MC."

Page 12 Line 21: "The relationship between mussel $\delta^{15}N$ and TDIN concentration within the estuary was much stronger when omitting site Cl and not significant when omitting site MC." |
| 44 | Page12:
21 Relationship can't be 'higher'. The r2 value can be higher, the relationship can be stronger etc. Though the slope of the line doesn't change much with removal of CI site, the fit improves. I mention earlier but you should also clarify if you keep the WO site in the regression. | We agree and will exchange the word "higher" with "stronger" and will also add the word "within the estuary" so that the sentence will read as follows: "The relationship between mussel $\delta^{15}N$ and TDIN concentration within the estuary was much stronger when omitting site Cl and not significant when omitting site MC." |
| 45 | Page12:
21b Good explanation of N cycling dynamics at this site. Could you include something similar for the MC site, even if it's conjectural it would be useful given how different the site was relative to the others. POM and mussel 15N are | We agree that we have to discuss site MC in more detail, even if it can only be speculative only. Rather than having a trend within the estuary, it could be that mussel isotope values are affected by different processes that are happening on a spatial scale within the estuary. This would blur a clear interpretation of the data. The two sites that strongly affect any relationship |

linked but nitrate 15N negatively linked to mussel 15N (driven by MC site). Could it be that POM sources are not within-estuary? If you're estuary is N-limited then production should be low, could be that POM is all sourced outside (upstream I imagine) and within-estuary nitrate 15N and nitrate concentrations aren't important to POM production. This could explain uncoupled 15N between POM and NO3. Do you have evidence of this? This would still be in line with the overall story here, reinforcing need for site-specific information and management approaches.

are Cl and MC, therefore the manuscript will include the following:

- Cl: A likely explanation for why Cl is different is already described in detail in the discussion (Page 12 Line 21)

- MC: We already discussed the idea that the low $\delta^{15}N$ of mussels at MC (and therefore the negative relationship with TDIN) is due to the fact that at higher nitrogen concentrations can lead to primary producers being choosier which leads to a negative relationship between nutrient concentration and mussel (Page 12 Line 1-20). We will further explore the idea that MC is different by adding (Page 12 Line 31): "Site MC was closest to the ocean, was one of the deepest sites and had a higher TDIN concentration compared to all other sites, which were not different with regard to TDIN concentration between them. This emphasises that the differences in mussel $\delta^{15}N$ between sites might be due to site-specific nutrient cycling processes in our estuary and might not reflect nitrogen pollution itself."

We would further like to thank the reviewer for their idea that POM is originating from outside the estuary (upstream). This is a very interesting speculation and we will add this into the discussion on page 12 Line 21as follows: "An alternative explanation would be that POM could originate upstream where nitrate might have had higher $\delta^{15}N$ values (not quantified in this study). Upon entering the estuary, POM mixes with estuarine POM, uncoupling the within-estuary $\delta^{15}N$ nitrate and POM $\delta^{15}N$ values. This could also explain the strong relationship between $\delta^{15}N$ in mussels and the distance from the estuary mouth found in our study. Such a strong relationship can be expected in estuaries with low pollution levels due to the aforementioned mixing, while little spatial variability in $\delta^{15}N$ values of primary consumers can be expected in heavily polluted estuaries due to the dominance of upstream POM, as was shown by Oczkowski et al. (2008)."

| 46 | Page 13 18 but your nitrogen sources of nitrate and ammonium were not different between sites (except MC) so this seems | We agree that differences in POM [15]N might drive differences in mussel [15]N and that this could be reflected in relationship between mussel [15]N and distance from the estuary mouth. We will therefore delete this section and |

| | | |
|---|---|---|
| | unlikely to explain differences in mussel 15N, no? More likely differences in POM 15N drove differences in mussel 15N and is reflected in relationship between mussel 15N and distance from mouth. It seems like there are other n cycling effects that are occurring here and could help to explain the negative (or lack of) correlation between 15N-NO3 and 15N-mussel (or TDIN and 15N-mussel. | incorporate parts of it earlier within the discussion, specifically where we will now discuss the strong relationship between mussel $^{15}$N and distance to estuary mouth (page 12 Line 2): "An alternative explanation would be that POM could originate upstream where nitrate might have had higher $\delta^{15}$N values (not quantified in this study). Upon entering the estuary, POM mixes with estuarine POM, uncoupling the within-estuary $\delta^{15}$N nitrate and POM $\delta^{15}$N values. This could also explain the strong relationship between $\delta^{15}$N in mussels and the distance from the estuary mouth found in our study. Such a strong relationship can be expected in estuaries with low pollution levels due to the aforementioned mixing, while little spatial variability in $\delta^{15}$N values of primary consumers can be expected in heavily polluted estuaries due to the dominance of upstream POM, as was shown by Oczkowski et al. (2008)." |
| 47 | Page 13 18 Your MC site may be influencing interpretation too much. If you had to interpret these data without the MC site, how would you do so? Does it change your overall conclusions? | We agree and will weaken the dependency of the discussion using relationships only. We will do this by

i) Acknowledging that the relationships are strongly driven by MC (see also our reply to your comment #38 & #43): - Page 9 Line 11: "The concentrations of total dissolved inorganic nitrogen …. were higher towards the estuary mouth (Fig. 2), although these relationships were weak …. and were driven by site MC only." - Page 11 Line 13: "In the Swan River estuary, NO$_3$ was enriched and there was a positive relationship between nitrate $\delta^{15}$N and the concentration of NO$_x$ throughout the estuary, although this was strongly driven by site MC." - Page 12 Line 7: "Our study showed a positive relationship between food (POM) and mussel $\delta^{15}$N, but a negative relationship between nitrate $\delta^{15}$N and consumers (mussels), which was strongly affected by site MC." - Page 12 Line 21: "The relationship between mussel $\delta^{15}$N and TDIN concentration within the estuary was much stronger when omitting site Cl and not significant when omitting site MC."

ii) Interpreting the data without site MC (Page 12 Line 31):" Site MC was closest to the ocean, was one of the deepest sites and had a higher TDIN |

concentration compared to all other sites, which were not different with regard to TDIN concentration between them. This emphasises that the differences in mussel $\delta^{15}N$ between sites might be due to site-specific nutrient cycling processes in our estuary and might not reflect nitrogen pollution itself."

This will not change our overall conclusion that mussels can be used as indicators for site-specific differences in pollution or nutrient cycling, which is "…important information for local management, but would have gone undetected at high pollution levels as the larger deviations of nitrogen stable isotope values would have made such small differences in mussel values invisible." (page 14 Line 8)

| 48 | Page 13 18 I would like to see more results regarding the POM and its connection to N-cycling. You have a fairly strong trend between 15N-mussel and distance from estuary mouth. What is driving this? | - We agree and will add more results for POM in the results section (3.4. Stable isotope values of POM). This will then read as: "POM $\delta^{15}N$ values were between 6.2 and 9.9 ‰ with no significant difference between sites ($F_{6,25}$= 1.327). A weak but significant negative relationship between POM $\delta^{15}N$ values and TDIN concentration was detected ($r^2$ =0.163, y = -0.044x + 9.37, $F_{1,28}$ = 5.44), while a significant positive relationship between nitrogen stable isotope signatures of POM and mussels was found ($r^2$ =0.303, y = 0.20x + 7.40, $F_{1,14}$ = 6.08) (Fig. 4 NEW). The relationship between $\delta^{15}N$ of POM and nitrate was not significant; however this calculation was based on only five data points where simultaneous measurements of the two $\delta^{15}N$ values were available, making this result arguable." |
| | | - We will also include the figure showing the positive relationship between mussel and POM $\delta^{15}N$ (new Figure 4) (please also see your comment #22) |
| | | - We will also discuss the strong trend between $^{15}N$ of mussel and distance from estuary mouth as follows:"…. An alternative explanation would be that POM could originate upstream where nitrate might have had higher $\delta^{15}N$ values (not quantified in this study). Upon entering the estuary, POM mixes with estuarine POM, uncoupling the within-estuary $\delta^{15}N$ nitrate and |

| | | POM $\delta^{15}N$ values. This could also explain the strong relationship between $\delta^{15}N$ in mussels and the distance from the estuary mouth found in our study. Such a strong relationship can be expected in estuaries with low pollution levels due to the aforementioned mixing, while little spatial variability in $\delta^{15}N$ values of primary consumers can be expected in heavily polluted estuaries due to the dominance of upstream POM, as was shown by Oczkowski et al. (2008)" |
|---|---|---|
| 49 | Page 14
- 4 correlated to nitrogen concentrations. But these were all negative correlations, no? | - We agree that this sentence might have been misleading for the reader. To avoid this we will delete "…that correlated to differences in nitrogen concentrations...". |
| 50 | Page 14
15N-mussel negatively correlated to 15N-NO3. 15N-NO3 positively (though MC weighs heavily) related to NOx concentration. High NO3 reflected in 15N-NO3 does not appear in 15N-mussels as sites with high 15N-NO3 and NO3 have low 15N-mussels, no? So mussels don't appear to be good indicators of NO3 sources as they don't reflect 15N-NO3, no? | - We agree with your comment that in our system mussels are not good indicators for NO3 sources. We believe that this is due to the fact that this estuary showed low nitrogen pollution during the study period (e.g., page 31 Line 31). However, because there are stable differences between sites, we like to argue that mussels are still good indicators for site specific nutrient cycling. Please also see our reply to your comment #5.

To reflect what we have just said, we will rewrite this section as follows: "The stable spatial differences in mussel $\delta^{15}N$ values over time highlight the value of this organism as a bioindicator of spatial water quality assessment. The negative trends between mussel $\delta^{15}N$ values and nitrate concentration or nitrate $\delta^{15}N$ values emphasize that mussels might not be good indicators for $NO_3$ sources in systems with low pollution levels. Instead, the small differences in mussel stable isotope signatures might reflect differences in site specific nutrient cycling caused by physicochemical conditions or biological factors rather than nitrogen pollution." |
| 51 | Page14:
In the discussion and conclusion sections you refer to mussels reflecting nitrate pollution but the link is weak, dependent on MC, and negative. Explain how these | We agree with this comment and will simplify or conclusion to: " The stable spatial differences in mussel $\delta^{15}N$ values over time highlight the value of this organism as a bioindicator of spatial water quality assessment. The negative trends between mussel $\delta^{15}N$ values and nitrate concentration |

| | |
|---|---|
| connections interact or simplify your message in the discussion and conclusion. The emphasis appears to be on nitrate but the linkages between nitrate and mussel tissue are unclear. | or nitrate $\delta^{15}N$ values emphasize that mussels might not be good indicators for $NO_3$ sources in systems with low pollution levels. Instead, the small differences in mussel stable isotope signatures might reflect differences in site specific nutrient cycling caused by physicochemical conditions or biological factors rather than nitrogen pollution." |

---

## Author Comment (AC2) · 9 Mar 2016

Dear Referee #2,

Thank you very much for your very helpful and positive comments on our manuscript "Can mussels be used as sentinel organisms for characterisation of pollution in urban water systems?" by E. S. Reichwaldt and A. Ghadouani (hess-2015-523). They provide very important feedback and improve the clarity of this manuscript.

We have now prepared replies to all of your comments and would here like to present how we will incorporate them into the next version of the manuscript.

We will first reply to your general comments, before we will discuss how we will address each of your specific comment in more detail.

**1) General comments**

| No. | Comments | Response |
|-----|----------|----------|
| 1 | [Structure]
The article is well written and structured. When going first through the manuscript, I had the impression that the introduction was pretty long (it is almost 0.25% of the text). Having said that, there is a lot of useful information and references included. One option could be to shorten a bit the introduction, or to introduce a few sub-headings in order to make it an easier read: basically it is about (1) increasing human impact on aquatic ecosystems, (2) the need for a better understanding of the spatial and temporal variability of pollution levels with a view to better manage these often irreversibly impacted systems, (3) the focus on nutrient pollution, (4) the use of stable isotopes (especially of N) for investigating anthropogenic nutrient pollution, and (5) the introduction of mussels as a sentinel organism in that specific context. | We agree that the introduction is long and will shorten it. We will i) delete part of the first paragraph (Page 2 Line 9 – 13) and ii) delete the first sentence of the second paragraph to make sure that we get to the point more quickly. By this, we hope to achieve a good balance between "getting to the point quickly" and "giving a broad picture of state of pollution management", which we think is appropriate for this journal that has such a large community of readers. |

**2) Specific comments**

| 2 | 1-Introduction [pages 2-4]: When reading the introduction, and more specifically the paragraphs to the end where mussels are introduce as sentinel organisms, I was surprised (unless I am mistaken) not to learn about what species have eventually been used for this study. I think this is a very important aspect that the authors have not taken into consideration for their manuscript. In an area where they expect living organisms to be a living archive of the local average environmental conditions it is essential to know a minimum about the metabolism of that organism. Especially in a journal that has a large community of readers from hydrological sciences, we cannot necessarily expect them to know much about this topic. Moreover, since this is a kind of proof-of-concept study, the authors should carefully describe the organisms, growth rates, sensitivity to changing environmental conditions etc. These aspects are likely to be crucial when it comes to eventually understand and discuss the isotopic signatures of N in the mussel's foot tissue. As mentioned further down in this assessment, there is existing literature in this respect and it would certainly be of value to take this into consideration in a revised version of the manuscript. | We agree and will include the species that we used (i.e. blue mussel, *Mytilus edulis*) in various places within the manuscript. The sentences will now read as follows: |
|---|---|---|
| | | Abstract: Page 1 Line 20: "The main aim of this study was to assess the suitability of nitrogen stable isotope as measured in mussels (*Mytilus edulis*), as an indicator able to resolve spatial and temporal variability of nitrogen pollution in an urban, tidally influenced estuary (Swan River estuary; Western Australia)." |
| | | Introduction: Page 4 Line 9: "Bivalves on the other hand, which include the blue mussel are primary consumers with limited movement, and have been suggested as suitable site-specific bioindicators of time-averaged persistence of nutrient pollutants, because their isotopic signature fluctuates less than that of their food sources due to longer tissue turnover rates (Raikow and Hamilton, 2001; Post, 2002; Fukumori et al., 2008; Fertig et al., 2010)." |
| | | Introduction: Page 4 Line 21: "The main aim of this study was to identify the variability of nitrogen concentration in an urban estuary over time and space and to ascertain the suitability of the isotopic signature ($\delta^{15}$N) of blue mussel (*Mytlius edulis*) tissue as an indicator of nitrogen pollution in urban water systems." |
| | | Materials and methods: Page 5 Line 19: "Seven sites within the Lower Swan River estuary were sampled 6 times for blue mussels and 9 times for nutrients,…" |
| | | Materials and Methods: Page 6 Line 10: "Nine blue mussels per site were randomly taken from the pylons of the jetties at each site from between 20 and 40 cm depth and brought into the laboratory on ice in bags containing |

| | | water from the respective site.: |
|---|---|---|
| | | We will further include a short paragraph to introduce this mussel species (Page 4 Line 13) in the introduction: "The blue mussel, *Mytilus edulis*, is a common sessile bivalve in estuarine and marine environments that is able to adapt to a wide range of environmental conditions, such as food concentration, temperature and salinity (e.g., Thompson and Bayne, 1974; Widdows et al., 1979; Zandee et al., 1980; Almadavillela, 1984), and that shows low sensitivity to anthropogenic pressures (Mainwaring et al., 2014). As such, this species is able to thrive at different pollution levels and has therefore been used as an indicator species for pollution (Phillips, 1976) and as a model organism for physiological, genetic and toxicological studies (Luedeking and Koehler, 2004) for some time." |
| 3 | 2-Material and methods [page 5 study sites & 6 sampling and analyses]: When reading the changing conditions in the Swan River estuary, one could expect differences between mussel species that are exposed to these fluctuations in salinity (between high tide and low tide). Is there only one mussel species in the studied area? If not (what is very likely), what are the other species that are present – what species has the sampling protocol been targeting – was it a mix of species – how sure can we be that different sensitivities to changing environmental conditions (including pollution) can lead to differences in metabolic activity? | We believe that this comment directly links to your previous comment #2 and by clarifying that we only used one species (blue mussel, *Mytilus edulis*) we believe this comment has been addressed by our previous reply. We would like to note that by using only one species we made sure that differences in metabolisms are restricted to within-species variability. |
| 4 | 3-Results [page 8 physicochemical parameters]: given that the study was carried out during rather dry conditions, the prevailing environmental parameters measured in the investigated area have also been rather unusual as stated in the manuscript. Here again, it would be interesting to see how the mussels populations have responded to that (if at all) – is there any information available on that? | We agree that conditions were unusually dry during our study. Unfortunately there is no previous data on this mussel population (e.g., abundance, physiology) that could be used for comparison with our study.

However, we would like to emphasise that blue mussels are known to adapt well to varying conditions (will be state in the new version of the manuscript on Page 4 Line 13) and that, because the mussels within the estuary all experienced the same conditions, these dry conditions will not |

| | | affect our conclusions. |
|---|---|---|
| 5 | 3-Results [page 8 physicochemical parameters]: On page 8, line 10 units should be added to salinity. | We would like to note that salinity does not have a unit as it is a ratio of the conductivity of a seawater sample and a standard potassium chloride solution (see UNESCO (1985): The international system of units (SI) in oceanography, UNESCO Technical Papers No. 45, IAPSO Pub. Sci. No. 32, Paris, France.) We will therefore not include a unit. |
| 6 | 3-Results [page 8 physicochemical parameters]: On page 10 the delta symbol should be homogenised. | We agree and will homogenised delta symbols by avoiding using them as a capital at the beginning of a sentence. |
| 7 | 4-Discussion [page 13]: In lines 6 to 8 I would be careful when stating that stable isotope signatures in mussels of tidally influenced estuaries are less impacted by seasonal changes in watershed input and chemistry compared to large rivers. This statement make sense considering the results of this study, but given the particularly dry conditions that prevailed during this investigation and the proof-of-concept character of this study, there need most probably to be more investigations before a strong statement in this sense. | We agree with this and have weakened this statement by restating it as follows: "Our results therefore indicate that while high seasonal variations of stable isotope signature in mussels can be connected to seasonal changes in watershed input and chemistry in large rivers (Fry and Allen, 2003), this is less pronounced in tidally influenced estuaries or during drier conditions with low freshwater input." |
| 8 | 5-Conclusion [page 13]: A similar comment as for the point above can be made for the 1st paragraph of the conclusion. | We agree and will rewrite the second sentence of the first paragraph as: "As such, stable isotope analysis of a model organism, such as the blue mussel can deliver essential information for future decentralised water management practices that are focused on local process understanding." |
| 9 | 5-Conclusion [page 13]:
Of interest could also be to see if there are differences in signatures between species. | We only analysed stable isotope signature of one species, blue mussel, which we now clarified throughout the manuscript as show in our reply to your comments #2 and #3. |
| 10 | 5-Conclusion [page 13]:
In the conclusion it is stated that the future studies should contribute in similar (low) polluted systems to better understand the baseline of spatial natural isotopic variability in urban aquatic systems. I was wondering if | We agree that this was misleading and we will therefore restate this sentence, which is an additional suggestion for future studies to gain a better understanding of systems with varying and partly low pollution levels. We will rewrite it as follows: "In addition, we advocate future studies in similarly (low) polluted systems that include stable isotope |

| | | |
|---|---|---|
| | this is not somehow contradictory with what is announced in the title – are mussels then really used in the sense of sentinels of pollution or rather as indicators of the baseline of 'spatial natural isotopic variability in urban aquatic systems'.

Again here I am possibly confused by the fact that no information is given on how sensitive those organisms are eventually to pollution. | analysis of other food web end-members and nutrients of the groundwater, to develop baselines of spatial natural isotopic variability in urban aquatic systems which will help identifying the importance of local biogeochemical processes for pollution control." We believe that this is reflected in the title.

We agree that this information has been missing and will include that blue mussels are not very sensitive to pollution by human activities. As such, this organism is able to thrive at different pollution levels indicating that their stable isotope signature should be an ideal indicator to identify differences in pollution levels. We will include this in the introduction (*Page 4 Line 13*): "The blue mussel, *Mytilus edulis*, is a common sessile bivalve in estuarine and marine environments that is able to adapt to a wide range of environmental conditions, such as food concentration, temperature and salinity (e.g., Thompson and Bayne, 1974; Widdows et al., 1979; Zandee et al., 1980; Almadavillela, 1984), and that shows low sensitivity to anthropogenic pressures (Mainwaring et al., 2014). As such, this species is able to thrive at different pollution levels and has therefore been used as an indicator species for pollution (Phillips, 1976) and as a model organism for physiological, genetic and toxicological studies (Luedeking and Koehler, 2004) for some time." |
| 11 | 5-Conclusion [page 13]:
As a last comment, one could also say that nutrient pollution is not really an urban problem or at least the origin of it can most of the time be found further upstream in agricultural parts of the catchments. In urban environments, one could also be targeting other sources of pollution, such as heavy metals, xenobiotics, etc. | We agree with this and we have mentioned its future application as sentinels for non-nutrient co-occurring pollutants (such as oils, heavy metals) in the abstract (last sentences), and the conclusion (Page 13 Line 22). |
| 12 | Concluding remarks: This manuscript is certainly a very interesting contribution for the readers of this journal and I enjoyed very much reading it. It is an interesting case study – or more specifically a proof-of-concept study – | We agree that using mussels as an indicator for pollution is not new and we will include citations for that (e.g. see our reply to your comment #2 and #10) to weaken the innovative aspect of this study. We further agree with you that the use of this approach in an urban, tidally influences estuary |

| | introducing mussels as a sentinel organism for investigating nutrient pollution in an urban aquatic environment. Since existing literature on similar applications/studies is not much referred to in the manuscript, the innovative character of this study might however be slightly overrated. | makes this study novel and interesting and that it presents a proof-of-concept study. |

---

## Author Response (AR1)

**HESS-2015-523 Revision report**
Can mussels be used as sentinel organisms for characterisation of pollution in urban water systems?
Elke S. Reichwaldt and Anas Ghadouani

Prof Dr Erwin Zehe
Chief-executive editor HESS
Institute of Water Resources and River Basin Management,
Karlsruhe Institute of Technology KIT
Kaiserstrasse 12
76129 Karlsruhe
Germany

Dear Dr Zehe,
Thank you and the two reviewers for providing insightful comments and feedback on our manuscript. They helped to greatly improve this manuscript. The main changes include a streamlined and clearer structure of the introduction and discussion; a discussion of the connection between POM and mussel $\delta^{15}$N supported by additional detailed POM data and a new Figure (Fig. 4); an introduction of our study organism (blue mussel); the inclusion of more recent publications on the use of mussels as indicators of pollution; and interpreting the data without site MC.
We have now completed the revisions and are happy to provide this detailed point-by-point revision report along with the revised manuscript for your consideration. We have highlighted the sections in the manuscript which have been amended or re-written.

| | **Editor** | | |
|---|---|---|---|
| | **Comment** | **Response** | **Location in text** |
| 1 | I very much agree with both reviewers that the method you proposed is interesting and has quite some innovation potential for assessment of pollutant concentration in estuaries. As such the paper is well within the scope of HESS and will optionally be published. | Thank you | - |
| 2 | I also agree with both reviewers that the manuscript | | |

| | | |
|---|---|---|
| would greatly benefit from | | |
| a) a clearer structure, | a) We have streamlined the introduction along the comments from the two reviewers and rearranged the discussion to fit the aims (introduction: reviewer 1 comments 2, 11, reviewer 2 comment 2; discussion: reviewer #1 comment 41. | a) Page 2 Line 9, Page 2 Line 14-23, Page 2 Line 15; Page 2 Line 25; Page 3 Line 33, Page 4 Line 5, Discussion |
| b) being more to the point with respect to terms like "pollution" and | b) we specified the pollution we are referring to throughout the manuscript | b) e.g., Page 1 Line 22; Page 2 Lines 26, 28, 30, 31, 32; Page 3 lines 1, 3, 7; |
| c) from providing references to more recent similar studies dealing with the potential of muscles as sentinel organisms for N-pollution. | c) We have now included three more recent references (Wang et al. 2013; Wen et al 2010; Fry et al 2011) that looked at mussels as indicators of nutrient pollution in lakes and estuaries and we compare their reported ranges in $\delta^{15}N$ of mussels with values in our study: | |
| | "In addition and identical to our study, the range of $\delta^{15}N$ values for nitrate and POM has been shown to be wider than the range for primary producers, indicating a time-averaging effect in mussels (Gustafson et al., 2007; Wang et al., 2013). Previous studies reported mussel $\delta^{15}N$ values between +6.6 and +16.7 ‰ in densely populated areas (Cabana and Rasmussen, 1996), polluted inland waterbodies (Wen et al., 2010; Wang et al., 2013) and a eutrophic estuary (Fry et al., 2011). Our values are at the lower end of this range, with | Page 12 Line 12 |

| | | mussel $\delta^{15}N$ values in our study being between 6.8 - 10.3 ‰, indicating the estuary is not highly polluted by wastewater, agriculture or fertilizers..” | |
| --- | --- | --- | --- |
| | | We have further added a recent publication on the use of other primary producers (non-mussels) as indicators of nutrient pollution to show the wide use of this approach (Xu and Zhang 2012).

Please, see also our reply to reviewer 1 comment 13 and reviewer 2 comment 12. | Page 3 Line 17 |
| 3 | Both reviewers also pointed out several substantial points that need to be clarified within the revised manuscript for fully explaining your findings and for evaluating to which extent they may be generalized.

a) This is for instance the joint effect of particulate organic matter and NO3, or | a) We agree that including more results on the POM and $NO_3$ is essential to better explain the results of our study. Reviewer 1 has commented on this and we have included all of their suggestions (comments 4, 22, 45, 46, 48) in the revised manuscript. For instance, we have included more results on POM (see 3.4 Stable isotope values of POM), a new Figure (Fig. 4) showing the relationship between mussel and POM $\delta^{15}N$ and we now also offer an alternative explanation for the uncoupling of δ15N of POM and nitrate. Please see specific comments for more details replies. | a) e.g., Figure 4; Page 9 Line 30; Page 10 Line 1; Page 13 Line 2 |

| | | | |
|---|---|---|---|
| | b) whether different species were included into the sampling protocol and whether their metabolism might react in a different manner to inherent dynamic changes of environmental conditions in an estuary. | b) Throughout the revised manuscript, we now clarified that we only included one species, i.e. the blue mussel, *Mytilus edulis*. As such, any difference in metabolisms is restricted to within-species variability. Please also see comments 2 and 3 from reviewer 2. | b) e.g.,  Page 1 Line 21, Page 4 Line 5, Page 4 Line 22, Page 5 Line 19, Page 6 Line 10, |

| | **Reviewer 1** | | |
|---|---|---|---|
| | **Comment** | **Response** | **Location in text** |
| 1 | I think the objectives could be improved | We agree and have restated the objectives using the suggestion from your comment #14. The objectives now read as follows: "Specifically, we anticipated that (1) a higher input of nitrogen-rich waters upstream would lead to a higher isotopic signatures of nitrate, (2) spatial differences in the level of nitrates in the water would lead to spatial differences in mussel isotopic signature, and (3) the increased distance from the mouth would lead to an elevated $^{15}$N values in mussels due to elevated $^{15}$N inputs from nitrogen-rich waters upstream. " | Page 4 Line 20 |
| 2 | Some of the introduction/discussion regarding water management approaches streamlined to move the reader more quickly to the meat of the paper. | We believe that this comment is a short version of comment #11 and we will reply in more detail below. In brief, we have deleted and rephrased sentences. By this, we believe that we have achieves a good balance between "getting to the point quickly" and "giving a broad picture of state of pollution management", which we think is appropriate for HESS that has such a large community of readers. | Page 2 Line 9, Page 2 Line 14-23, Page 2 Line 15, Page 2 Line 25, Page 3 Line 33 |

| | | | |
|---|---|---|---|
| 3 | I was a bit confused by exactly what was meant by pollution – nitrate or nitrogen generally. There is a strong focus on nitrate but the results don't point strongly to mussels reflecting nitrate concentration or 15N composition and therefore a larger focus on the N-cycle may be needed to explain the results observed here. | We agree that this needed clarification. The analysis of the nitrogen signature in general has proven to be a powerful tool as an indicator of anthropogenic contamination. Our study looks specifically at the stable isotope signature of nitrate. Then, to additionally test if mussels can be used as bioindicators of nitrogen pollution, we broaden our objectives and look at nitrogen. | |
| | | We made this clear throughout the manuscript by exchanging the word "nutrients" with "nitrogen" where appropriate | e.g., Page 1 Line 22; Page 2 Lines 26, 28, 30, 31, 32; Page 3 lines 1, 3, 7; |
| | | In addition, we restated the objectives: "The main aim of this study was to identify the variability of nitrogen concentration in an urban estuary over time and space and to ascertain the suitability of the isotopic signature ($\delta^{15}N$) of mussel tissue as an indicator of nitrogen pollution in urban water systems. Specifically, we anticipated that (1) a higher input of nitrogen-rich waters upstream would lead to a higher isotopic signatures of nitrate, (2) spatial differences in the level of nitrates in the water would lead to spatial differences in mussel isotopic signature, and (3) the increased distance from the mouth would lead to an elevated $^{15}N$ values in mussels due to elevated $^{15}N$ inputs from nitrogen-rich waters upstream. | Page 4 Line 20 |
| 4 | More attention should be paid to POM and how/why or whether POM is decoupled from NO3 and how this | We agree and we have now included the following paragraph: "An alternative explanation would be | Page 13 Line 2 |

| | | | |
|---|---|---|---|
| | relates to the 15N of the mussels. Your strongest figure is 15N-mussel vs the distance from estuary (some of others are strongly influenced by one site, MC) and this is not fully explained in the discussion. | that POM could originate upstream where nitrate might have had higher $\delta^{15}N$ values (not quantified in this study). Upon entering the estuary, POM mixes with estuarine POM, uncoupling the within-estuary $\delta^{15}N$ nitrate and POM $\delta^{15}N$ values. This could also explain the strong relationship between $\delta^{15}N$ in mussels and the distance from the estuary mouth found in our study. Such a strong relationship can be expected in estuaries with low pollution levels due to the aforementioned mixing, while little spatial variability in $\delta^{15}N$ values of primary consumers can be expected in heavily polluted estuaries due to the dominance of upstream POM, as was shown by Oczkowski et al. (2008)." | |
| 5 | There is a strong emphasis on site-specific characteristics influencing mussels however, aside from MC, the concentration of NO3 and NH4 were fairly low and not correlated across sites. So, NO3 and NH4 not likely to explain site-specific 15N-mussel variability. This should be addressed quickly in the discussion section. | We agree with the reviewer. Because $NO_3$, $NH_4$, or TDIN concentrations are very similar for many of the sites they cannot completely explain $\delta^{15}N$ variability in mussels. We will include statements about this within

i) the discussion:" Site MC was closest to the ocean, was one of the deepest sites and had a higher TDIN concentration compared to all other sites, which in turn did not show differences in TDIN concentrations between them. This emphasises that the differences in mussel $\delta^{15}N$ between sites detected in our estuary might rather reflect site-specific nutrient cycling processes than nitrogen pollution itself." | i) Page 13 Line 21 |

| | | | |
|---|---|---|---|
| | | ii) the conclusion:" The negative trends between mussel $\delta^{15}N$ values and nitrate concentration or nitrate $\delta^{15}N$ values emphasize that mussels might not be good indicators for $NO_3$ sources in systems with low pollution levels. Instead, the small differences in mussel stable isotope signatures might reflect differences in site specific nutrient cycling caused by physicochemical conditions or biological factors rather than nitrogen pollution." Please also see our reply to your comment #50. | ii) Page 14 Line 22 |
| 6 | Page 1: 19 higher nitrogen stable isotope signature. Enriched in 15N is more accurate. Purely preference here, you did well | We agree that both phrases can be used. We prefer to keep "higher nitrogen stable isotope signature" in the abstract to make it easier for readers who are not entirely familiar with stable isotope jargon. | - |
| 7 | Page 1: 23 Can you omit the sentence beginning with "Our results showed a trend. . ." I think the sentence isn't necessary in the abstract. | We agree and deleted this sentence in the abstract. | Page 1 |
| 8 | Page 1: 26 What are natural values? Maybe state within range of observed values within estuaries of W. Australia | We agree that this was ambiguous. We clarified it by rewriting this sentence to :" … nitrogen stable isotope values of nitrate throughout the estuary were well within natural values of uncontaminated groundwater, organic nitrate from soils or marine derived sources, indicating groundwater inflow rather than pollution by human activity was responsible for differences between sites."

The natural values that we refer to are now also mentioned in more detail with relevant references in the following sections | Page 1 Line 23 |

| | | i) the discussion:" Because the isotopic signatures of nitrates were well in the range of values reported for surface water (~ -4 - +9 ‰; Xue et al., 2009), uncontaminated groundwater (~ -1 - +8 ‰; Xue et al., 2009), organic nitrate from soils (0 - +10 ‰; Heaton, 1986), pristine streams (+1.8- +2.2 ‰; Harrington et al., 1998), or naturally available marine-derived dissolved inorganic nitrogen (c. 6-8 ‰; Dudley and Shima, 2010), our study does not suggest differences in the level of human impact between sites. Additionally, nitrate $\delta^{18}O$ values in our study are similar to values indicative of the atmospheric source (+20 - +80 ‰; Kendall, 1998; Xue et al., 2009), suggesting that the higher concentration and enriched signature of $NO_x$ at site MC is unlikely to result from anthropogenic pollution, but might rather be due to addition of $NO_x$ by groundwater inflow, potentially in combination with different productivity or biochemical processes at this site compared to any of the other sites. | i) Page 11 Line 17 |
| | | ii) the conclusion: " …were due to a natural input of nitrate uncontaminated groundwater (Xue et al., 2009) rather than human pollution.".

We are not aware of any stable isotope values for estuaries in Western Australia. | ii) Page 14 Line 19 |

| 9 | Page 1:
28 Delete " which allowed for the detection of spatial difference" | We agree and deleted this part of the sentence. | Page 1 Line 28 |
|---|---|---|---|
| 10 | Page 1:
29 change to 'organisms' | We agree and corrected this. | Page 1 Line 29 |
| 11 | Page 2:
First paragraph doesn't relate well to abstract or title. I would introduce this paper with the current state of affairs regarding nitrogen in urban water systems, then identify the problem – the quantification of a spatial and temporally varying regulated chemical species (nitrogen).
Page 2:
Second paragraph starts from the point of restoration and then proceeds to the problem – limited understanding of temporal and spatial variability of pollution (I would state nitrogen here, it is your focus) I suggest deleting most of paragraph 1 and improving paragraph 2 to more concisely state your research problem, question etc. Get to the point of the paper very quickly. | We agree that the introduction was too long and have shortened and streamlined it by,

i) deleting the third sentence of the first paragraph ("The high percentage….).

ii) deleting the last two sentences of the first paragraph ("In an attempt to reconnect….").

iii) deleting the first sentence of the second paragraph to make sure that we get to the point more quickly ("Typically the success rate…")

iv) deleting part of the third paragraph ("and will be even more impacted in the future. Nutrient pollution is of particular concern in many waterbodies, because…")

v) being more specific by exchanging the words "nutrient" and "pollutant" with "nitrogen" in the third paragraph.

vi) including a paragraph on our study organisms (*Mytilus edulis*, blue mussel) and its use as an indicator species. | i) Page 2 Line 9

ii) Page 2 Line 14-23

iii) Page 2 Line 15

iv) Page 2 Line 25

v) e.g., Page 1 Line 22, Page 2 Lines 26, 28, 30, 31, 32; Page 3 lines 1, 3, 7;

vi) Page 3 Line 33 – Page 4 Line 12 |

| | | By this, we believe that we have achieves a good balance between "getting to the point quickly" and "giving a broad picture of state of pollution management", which we think is appropriate for HESS that has such a large community of readers. | |
|---|---|---|---|
| 12 | Page 3:
11 – delete s from 'urbans' | We agree and corrected it. | Page 2 Line 27 |
| 13 | Page 4:
18 Citation for this? | We now added citations, and combine the two sentences. It now read as: "Earlier studies in polluted freshwater and marine systems found positive relationships between the concentration of nitrogen and the isotopic signature of nitrogen in mussels, and between the isotopic signature of nitrate-N and that of mussels (Cabana and Rasmussen, 1996; McClelland et al., 1997; Costanzo et al., 2001; Anderson and Cabana, 2005; Gustafson et al., 2007; Wen et al., 2010), suggesting that bivalves are suitable indicators of changes in nutrient pollution load from agriculture and wastewater to waterbodies." | Page 4 Line 12 |
| | Would be useful to reader to know how work in polluted waterbodies then relates to concentrations and isotopic composition presented in this paper – were polluted waterbodies exhibiting higher concentrations and heavier 15N values? Over a larger range? Etc | As suggested, we added a comparison between nitrogen concentration / $\delta^{15}N$ values found in our study and the values reported from polluted systems:

"In addition and identical to our study, the range of $\delta^{15}N$ values for nitrate and POM has been shown to be wider than the range for primary producers, indicating a time-averaging effect in mussels | Page 12 Line 12 |

| | | (Gustafson et al., 2007; Wang et al., 2013). Previous studies reported mussel $\delta^{15}N$ values between +6.6 and +16.7 ‰ in densely populated areas (Cabana and Rasmussen, 1996), polluted inland waterbodies (Wen et al., 2010; Wang et al., 2013) and a eutrophic estuary (Fry et al., 2011)." | |
| | | "Because the isotopic signatures of nitrates were well in the range of values reported for surface water (~ -4 - +9 ‰; Xue et al., 2009), uncontaminated groundwater (~ -1 - +8 ‰; Xue et al., 2009), organic nitrate from soils (0 - +10 ‰; Heaton, 1986), pristine streams (+1.8- +2.2 ‰; Harrington et al., 1998), or naturally available marine-derived dissolved inorganic nitrogen (c. 6-8 ‰; Dudley and Shima, 2010), our study does not suggest differences in the level of human impact between sites. Additionally, nitrate $\delta^{18}O$ values in our study are similar to values indicative of the atmospheric source (+20 - +80 ‰; Kendall, 1998; Xue et al., 2009), suggesting that the higher concentration and enriched signature of $NO_x$ at site MC is unlikely to result from anthropogenic pollution, but might rather be due to addition of $NO_x$ by groundwater inflow, potentially in combination with different productivity or biochemical processes at this site compared to any of the other sites" | Page 11 Line 17 |
| 14 | Page 4: Make sure objectives use same phrasing. "Would lead to. . ." is good and used in 2 of 3. Keep it uniform to | We agree with his comment and rephrased objective (2) as follows: "(2) spatial differences in the level of nitrates in the water would lead to spatial | Page 4 Line 24 |

| | | | |
|---|---|---|---|
| | help the reader. (2) is more of a conclusion | differences in mussel isotopic signature" | |
| 15 | Page 4:
26 change to 'nitrogen-rich' | We agree and changed it. | Page 4 Line 23 |
| 16 | Page 4:
27 '(2) distinct spatial difference in mussels. . .' This doesn't quite make sense. Do you mean to say that the number of mussels relates to the nitrate concentration? Or that the 15N composition of mussels reflect observed composition in nitrate. | We agree that this was expressed ambiguously. We restated objective 2 to "(2) spatial differences in the level of nitrates in the water would lead to spatial differences in mussel isotopic signature" to make this explicit. [see also your comment #14] | Page 4 Line 24 |
| 17 | Page 4:
29 'lead to increased anthropogenic signal'. Rephrase, you anticipate that you will observe elevated 15N due to elevated 15N inputs from nitrogen-rich waters, which follows your prediction (1). | We agree and rephrased objective 3 as follows: "(3) the increased distance from the estuary mouth would lead to elevated $^{15}$N values in mussels due to elevated $^{15}$N inputs from nitrogen-rich waters upstream." | Page 4 Line 26 |
| 18 | Page 5
8 Change to - prone to 'nutrient' pollution | We agree and added the word "nutrient" in the sentence. | Page 5 Line 6 |
| 19 | Page 6
General – clarify that the 15N composition is reported in units relative to an international standard (air usually). I assume the standard is the same for both isotope facilities used in this paper. Report it and clarify that the 15N concentrations you report are relative to the standard and are not absolute concentrations (isotope scientists know this, others may not). Same for 18O. This should be done in the methods section at a minimum, often re-stated in data tables as part of the units of 15N | We agree that this needs clarification. We can confirm that both institutions used the same standards ($\delta$ $^{15}$N: air; $\delta^{18}$O: Vienna Standard Mean Ocean Water, VSMOW) and that all values are reported in per mil (‰) with respect to the international standards.
We now include the following sentences in the method section:
- nitrate: "All values are reported in per mill (‰) with respect to the international standards ($\delta$ $^{15}$N: air; $\delta^{18}$O: Vienna Standard Mean Ocean Water, VSMOW)."
- mussels and POM: "All values are reported in per mill (‰) with respect to the international standard (air)." | Page 6 Line 25

Page 7 Line 17 |

| 20 | Page 7 1 Change to "To determine the isotopic composition of nitrogen in particulate organic matter (POM), a source of food for mussels, 0.7 – 2.5 L. . .." Avoid using 'signature' unless you've determined that the isotopic composition of POM is unique, particularly if you're only using one isotope for characterization. | We agree and changed the sentence as suggested to "To determine the isotopic composition of nitrogen in particulate organic matter (POM), which is the food source for mussels that presents the direct link between nitrate and the mussels, 0.7 – 2.5 L…". | Page 6 Line 31 |
|---|---|---|---|
| 21 | Page 7 1b Your hypothesis #2 is that mussel 15N corresponds to nitrate 15N, no? But here you say that mussels feed on POM so the reader is confused by the nitrate 15N hypothesis. You should rectify this earlier in the introduction somehow. Either focus on POM or state how N cycling would link nitrate and POM 15N composition. | We agree that we have made the link between $^{15}$N of nitrate, POM and mussels not clear enough. We now explain this in more detail in the introduction and have added the following sentences: "This signal is then passed on to higher trophic levels up the food chain (Cabana and Rasmussen, 1994; Carvalho et al., 2015): Elevated $\delta^{15}$N signals in nitrate have been shown to lead to elevated $\delta^{15}$N signals in organisms that directly take up nitrate from the water, such as phytoplankton and microbes (Harrington et al., 1998). These organisms form an important part of particulate organic matter (POM), which serves as food for filter feeders (e.g., mussels). Mussels that ingest POM with elevated $\delta^{15}$N signal will then also show a higher $\delta^{15}$N signal." | Page 3 Line 21 |
| | | "To determine the isotopic composition of nitrogen in particulate organic matter (POM), which is the food source for mussels that presents the direct link between nitrate and the mussels,…" | Page 6 Line 31 |

| 22 | Page 7
1c You state that mussel 15N and POM 15N are linked but you don't show in a figure. And the link between 15N POM and 15N NO3 is also not discussed in the results. | We believe that this comment refers to Page 9. We agree and have now included results for these two relationships:

i) we added a figure (Fig. 4) showing the significant, positive relationship between mussel and POM $\delta^{15}$N. We think it will be helpful for the reader, because in the revised manuscript we discuss in more detail the relationship between POM and mussel nitrogen stable isotope values. (see also your comment #48)

ii) We have further included a sentence on the relationship between $\delta^{15}$N of POM and nitrate in the revised manuscript. This reads as follows: "The relationship between $\delta^{15}$N of POM and nitrate was not significant; however as this calculation was based on only five data points where simultaneous measurements of the two $\delta^{15}$N values were available, the value of this result is uncertain." | i) Figure 4
Reference in text on Page 10 Line 1

ii) Page 10 Line 1 |
|---|---|---|---|
| 23 | Page 7
4 Change to "Harvested mussels were measured and dissected to obtain the foot tissue. . ." | We agree and changed it as suggested. | Page 7 Line 2 |
| 24 | Page 7
6 Was the foot tissue homogenized before isotope analysis or was the entire sample of 3 combined foot tissue used in the mass-spectrometer? If the entire sample was used, state so, if the sample was fully homogenized with mortar/pestle state that. As it is it seems there were 3 distinct pieces of foot tissue were dried together. | We agree that the description was unclear and have consequently changed it to:
"The feet of three individuals per site were combined, dried at 60°C for at least 24 h, fully homogenized with mortar/pestle, and stored in a desiccator until a subsample was analysed for mussel $\delta^{15}$N and C:N ratio." | Page 7 Line 3 |

| 25 | Page 8:
4 long term average based on how many years? Citation? | We have added the number of years the average was based on and the citation. The sentence now reads:
"Rainfall was below average in 2010 with 421 mm for the entire sampling period, while the average for this period was 690 mm in the previous 17 years (1993-2009; Bureau of Meteorology, 2016)." | Page 8 Line 3 |
|---|---|---|---|
| 26 | Page 8:
7 The comparison is between discharge during the winter of 2010 and the winter of 1994 and the conclusion is that 2010 discharge was lower than usual. Is there a published mean discharge value you can compare to? Or is the discharge of '94 the only published value for comparison? To state discharge is lower than usual you should have an average or trend of some sort for comparison. | We agree with this comment and no report the average discharge for 1993-2009 and the minimum and maximum values within this period. These values are taken from the Department of Water data base (included as a reference now). It now reads as follows:
"This resulted in a lower than usual discharge from the tributaries into the estuary with a mean discharge from the Swan River of $1.2 \times 10^5$ m$^3$ d$^{-1}$ in 2010 compared to an average discharge of $8.4 \times 10^6$ m$^3$ d$^{-1}$ for the period of 1993-2009 for the same season (min. – max: $1.99 \times 10^6$ m$^3$ d$^{-1}$ (2002) – $2.21 \times 10^7$ m$^3$ d$^{-1}$ (1996) (Department of Water, 2016)." | Page 8 Line 5 |
| 27 | Page 8:
10 Unusually high salinity? Is this relative to a published average salinity value for the estuary? Need citation or cleaner text. Either state the salinity was high throughout the area or high relative to a specific mean value (with citation if possible).

10b What are the units for salinity? I suggest adding the salinity recorded for the ocean water in the nearby area (or salinity of ocean water generally) for the reader to | We agree and now cite a previous study, which reports on salinity in this estuary. We will also add that seawater has a salinity of 35. The section will now read:
"This might have contributed to higher salinities throughout the entire estuary during this study than previously reported (Stephens and Imberger 1997) and no relationship between salinity and distance to the estuary mouth was detected. During high tide, the salinity at all sites was between 24.2 and 32.4 | Page 8 Line 9 |

| | | and there was no difference in salinity between sites which can be considered brackish to saline (salinity of seawater is 35)." | |
| --- | --- | --- | --- |
| | | We believe that this information together with the description of the Swan River Estuary (2.1 Study site) will now be sufficient to understand the dynamics of salinity in this estuary. | Page 5 Line 7-18 |
| | | We would like to mention that salinity does not have a unit as it is a ratio of the conductivity of a seawater sample and a standard potassium chloride solution (see UNESCO (1985): The international system of units (SI) in oceanography, UNESCO Technical Papers No. 45, IAPSO Pub. Sci. No. 32, Paris, France.) We will therefore not include a unit. | - |
| 28 | Page 8:
31 Change to "while nitrogen from NH4+ was greater at all other sites (Fig. 2)". | We agree and have changed it as suggested. | Page 9 Line 1 |
| 29 | Page 8:
31 Can omit sentence starting with "This is supported by significant. . .". It doesn't add much value compared to previous sentence. | We agree and have deleted the sentences. This section now reads as:
"On average, $NO_x$ was the dominant N source at MC, SCC and WO, while nitrogen from $NH_4^+$ was greater at all other sites (Fig. 2) (Kruskal Wallis one-way ANOVA, H = 59.0, df = 6). | Page 8 Line 31 |
| 30 | Page 9:
4 change to "The TN:TP ratio (weight) was between 0 and 6.5, with 84% of the ratios (by site? ) below 2.2". Move the rest of the paragraph to appropriate place in discussion OR condense to simple sentence that cites published thresholds for determining nitrogen | We agree and we have changed this to:
"The TN:TP ratio (weight) of particulate organic matter was between 0 and 6.5 with 84% of the samples in our study being below 2.2, indicating a high possibility of nitrogen limitation in this system (Redfield 1958; Geider and La Roche, 2002). | Page 9 Line 10 |

| | | | |
|---|---|---|---|
| | limitation (7.2 or 2.2). | | |
| 31 | Page 9:
18 "Analysis of stable isotope composition of NO3. . ." Change 'signature' throughout unless you're really talking about the uniqueness of a component's isotopic composition. | We fully agree and have exchanged "signature" with "value" or "composition" throughout the manuscript where appropriate. | e.g., Page 1 Line 24; Page 4 Line 26; Page 6 Line 25; Page 7 Line 17; Page 9 Line 14, 27; Page 10 Line 3, 7, 17; Page 13 Line 1, 8. |
| 32 | Page 9:
19 restate minimum concentration requirements | As suggested we now restated the minimum concentration requirement. It now reads as: "Analysis of the stable isotope composition of $NO_3$ was limited to a total of 25 samples that fulfilled nutrient concentration requirements for the analysis (0.71 µM $NO_3$-N)." | Page 9 Line 15 |
| 33 | Page 10:
1 Clarify sentence findings – I understood that POM 15N and mussel 15N collected at each site had a significant, positive relationship to one another. By fractionation effect of 0.6 do you mean that mussel 15N composition was on average 0.6 greater than POM 15N composition at same site? Clarify this for the reader, particularly if you're not including a figure. | We agree and decided to delete the part about the fractionation, as it is not relevant for the main message and conclusion of the paper. | Page 10 Line 3 |
| 34 | Page 10:
5 Move this sentence second in the paragraph. Move second sentence to the first sentence position. | We agree and changed the position of these two sentences. | Page 10 Line 7 |
| 35 | Page 10:
7 'smaller than range seen in 15N nitrate' (. . . to . . .) restate range of nitrate 15N to make it easier for the reader to compare the relative ranges of each. . | We agree and have changed this to:
"Values of $\delta^{15}N$ of mussels varied between 6.8 and 10.3 ‰ and the range was therefore smaller than the range seen in nitrate $\delta^{15}N$ (-1.3 and 10.4 ‰)." | Page 10 Line 7 |
| 36 | Page 10:
8 use lower case , not _. It would be better to rephrase the sentence so you are not starting with a greek letter. | We agree and restated the sentence. It now reads as:
"Mussel $\delta^{15}N$ was significantly different between sites (one-way ANOVA; $\delta^{15}N$: $F_{6,98} = 42.53$) and | Page 10 Line 10 |

| | | was negatively correlated with the concentration of total dissolved inorganic nitrogen ($r^2 = 0.486$, $F_{1,5} = 4.73$, $P < 0.1$) (Fig. 6)." | |
|---|---|---|---|
| 37 | Page 10:
9 "no temporal trend" sentence starts with a non-trend and ends with a significant (?) trend between 15N and distance to estuarine mouth, connect the two clauses with a 'though'. | We believe that this comment refers to the following (original) sentences (page 10 Lines 7-10): "No temporal trend in mussel $\delta^{15}$N was detected (Fig. 4). $\Delta^{15}$N of mussels was significantly different between sites (one-way ANOVA; $\delta^{15}$N: $F_{6,98} =$ 42.53) (Fig. 5) and mussel $\delta^{15}$N increased with increasing distance from the estuary mouth (Fig. 6)."

We have changed the second sentence as suggested in the previous comment (#36). We have also added that the increase with distance from the estuary mouth was significant. This section now reads as follows:
"No significant relationship between mussel length and mussel $\delta^{15}$N (linear regression; $F_{1,13} = 2.235$) and no temporal trend in mussel $\delta^{15}$N was detected (Fig. 5). Mussel $\delta^{15}$N was significantly different between sites (one-way ANOVA; $\delta^{15}$N: $F_{6,98} =$ 42.53) and was negatively correlated with the concentration of total dissolved inorganic nitrogen ($r^2 = 0.486$, $F_{1,5} = 4.73$, $P < 0.1$) (Fig. 6). When site Cl was omitted, the strength of the relationship increased ($r^2 = 0.838$, $F_{1,4} = 20.69$, $P < 0.05$), while the relationship was not significant with an $r^2$ of 0.009 only when site MC was omitted (Fig. 6). Mussel $\delta^{15}$N increased significantly with distance from the estuary mouth ($r^2 = 0.563$, y = 0.12x+7.74, | Page 10 Line 8 |

| | | | |
|---|---|---|---|
| | | $F_{1,110} = 141.65$) (Fig. 7) and showed a significant negative relationship between the $\delta^{15}N$ values of mussel and nitrate ($r^2 = 0.711$, $F_{2,10} = 24.65$) (Fig. 8)" | |
| 38 | Page 10:
Figures 3, 5 and 7 all strongly influenced by MC site. | We agree and now report and discuss the strong influence of MC on relationships at various places throughout the manuscript, including: | |
| | | "The concentrations of total dissolved inorganic nitrogen …. were higher towards the estuary mouth (Fig. 2), although these relationships were weak …. and were driven by site MC only." | Page 9 Line 5 |
| | | "In the Swan River estuary, $NO_3$ was enriched and there was a positive relationship between nitrate $\delta^{15}N$ and the concentration of $NO_x$ throughout the estuary, although this was strongly driven by site MC." | Page 11 Line 15 |
| | | "We also found a positive relationship between food (POM) and mussel $\delta^{15}N$, but a negative relationship between nitrate $\delta^{15}N$ and consumers (mussels), which was strongly affected by site MC." | Page 12 Line 20 |
| | | "The relationship between mussel $\delta^{15}N$ and TDIN concentration within the estuary was much stronger when omitting site Cl and not significant when omitting site MC." | Page 13 Line 11 |
| | | In addition, we included a paragraph in which we | |

| | | interpret the data without site MC: | |
|---|---|---|---|
| | | " Site MC was closest to the ocean, was one of the deepest sites and had a higher TDIN concentration compared to all other sites, which in turn did not show differences in TDIN concentrations between them. This emphasises that the differences in mussel $\delta^{15}$N between sites detected in our estuary might rather reflect site-specific nutrient cycling processes than nitrogen pollution itself." | Page 13 Line 21 |
| | | We would like to emphasize that, this does not change our overall conclusion that mussels can be used as indicators for site-specific differences in pollution or nutrient cycling. | |
| 39 | Page 10: Figure 5. You show scenarios with and without CI or MC sites, was WO site included in regressions? | We did not include the marine site (WO) in the regression. We agree that this has not been described clearly and we now included the following sentence in the figure legend: "WO was not included in the regressions." | Figure legend 6; page 28 Line 8 |
| | | There are two reasons why we did not include WO in the regressions: 1) the N-cycle is likely to be different in the estuary compared to the marine environment; 2) The purpose of this paper is to identify if mussels can be used as bioindicators within a system, which would be the estuary in our case. As such, including the marine site is not relevant but would rather confound trends and findings. The purpose of showing WO is purely to provide a baseline for a marine environment. | |

| 40 | Page 10:
31 avoid using 'site-specific' twice in same sentence. Restructure | We agree and substituted the second "site-specific" by "spatial". | Page 11 Line 1 |
|---|---|---|---|
| 41 | Page 11
It would be easier for the reader if the discussion followed directly from the 3 objectives stated in the introduction – nitrogen and 15N conc upstream; 15N mussel by site and nitrate conc; distance from mouth = anth signal. | We agree and by adopting changes suggested in this and the following comment (#42), the discussion of our data is now structured as follows:
1) nitrogen concentrations in the estuary (spatial; upstream/downstream)
2) discussion of nitrate $\delta^{15}N$ values (site specific; processes that lead to differences between these values).
3) Mussel $\delta^{15}N$ between sites and relationship between nutrient concentrations
4) Mussels and distance from estuary mouth
5) Mussel $\delta^{15}N$ over time and suitability as indicators | Discussion section |
| 42 | Page 11:
24 What do you mean by this sentence. Expand more. How does the fraction of NOx in the DIN pool explain site-specific variation in 15N? It's stated here but the reader doesn't understand how simply from the sentence | We agree that this paragraph was unclear and needed expanding. We restated it as follows:
"The fraction of NOx of the TDIN pool (%) was significantly different between sites (data not shown; $y = 0.15x-6.9$, $r^2 = 0.215$, $F_{1,23} = 6.30$, $P < 0.05$), with site MC having a higher mean fraction (mean $= 62.5\%$) compared to all other sites, except for SCC. An earlier study by Sugimoto et al. (2009) also found a positive relationship between nitrate $\delta^{15}N$ values and the nitrate fraction in TDIN which they explained by *in situ* isotopic effects during nitrification. However whether higher $\delta^{15}N$ values of nitrate at MC are related to site specific nitrification rates in our estuary needs further investigation, because the $\delta^{18}O$ and $\delta^{15}N$ values of | Page 11 Line 31 |

| | | nitrate are rather representative of atmospheric $NO_3$ deposition values (Durka et al., 1994; Fang et al., 2011) and nitrification is likely to play a minor role at ammonium concentrations <5 µM (Day et al., 1989) that prevail in the Swan River estuary." | |
|---|---|---|---|
| 43 | Page 12
First two sentences are redundant, simplify and merge. Sentence 1 is cumbersome with overuse of "15N values". Trend between mussel 15N and nitrate 15N strongly driven by site MC. As is relationship with TDIN. Without MC site, there is little to no trend. You should address this head-on in your discussion section. | We agree and restated these two sentence as follows:
"Earlier studies found that nitrogen $\delta^{15}N$ values are reflected in higher trophic levels in a predictable way with a positive relationship between $\delta^{15}N$ of nitrate, primary producer and primary consumer (e.g., mussels) (Cabana et al., 1994; Cabana and Rasmussen, 1996; Harrington et al., 1998; Oczkowski et al., 2008; Carvalho et al., 2015)."

We further discuss the fact that the trends are strongly driven by site MC throughout the manuscript (please also see our reply to your comment #38):

"The concentrations of total dissolved inorganic nitrogen …. were higher towards the estuary mouth (Fig. 2), although these relationships were weak …. and were driven by site MC only."

"In the Swan River estuary, $NO_3$ was enriched and there was a positive relationship between nitrate $\delta^{15}N$ and the concentration of $NO_x$ throughout the estuary, although this was strongly driven by site MC." | Page 12 Line 9

Page 9 Line 5

Page 11 Line 15 |

| | | "We also found a positive relationship between food (POM) and mussel $\delta^{15}$N, but a negative relationship between nitrate $\delta^{15}$N and consumers (mussels), which was strongly affected by site MC." | Page 12 Line 20 |
|---|---|---|---|
| | | "The relationship between mussel $\delta^{15}$N and TDIN concentration within the estuary was much stronger when omitting site Cl and not significant when omitting site MC." | Page 13 Line 11 |
| 44 | Page12:
21 Relationship can't be 'higher'. The r2 value can be higher, the relationship can be stronger etc. Though the slope of the line doesn't change much with removal of CI site, the fit improves. I mention earlier but you should also clarify if you keep the WO site in the regression. | We agree and exchanged the word "higher" with "stronger" and in addition add the word "within the estuary" so that the sentence will read as follows: "The relationship between mussel $\delta^{15}$N and TDIN concentration within the estuary was much stronger when omitting site Cl and not significant when omitting site MC." | Page 13 Line 11 |
| 45 | Page12:
21b Good explanation of N cycling dynamics at this site. Could you include something similar for the MC site, even if it's conjectural it would be useful given how different the site was relative to the others. POM and mussel 15N are linked but nitrate 15N negatively linked to mussel 15N (driven by MC site). Could it be that POM sources are not within-estuary? If you're estuary is N-limited then production should be low, could be that POM is all sourced outside (upstream I imagine) and within-estuary nitrate 15N and nitrate concentrations aren't important to POM production. This could explain uncoupled 15N between POM and NO3. Do you have evidence of this? This would still be | We agree that we have to discuss site MC in more detail, even if it can only be speculative only. Rather than having a trend within the estuary, it could be that mussel isotope values are affected by different processes that are happening on a spatial scale within the estuary. This would blur a clear interpretation of the data. The two sites that strongly affect any relationship are Cl and MC. To make this clear, we now included:

Site Cl:
A likely explanation for why Cl is different is described in detail in the discussion. | Page 13 Line 12 |

| | | | |
|---|---|---|---|
| | in line with the overall story here, reinforcing need for site-specific information and management approaches. | Site MC:
We discuss that the low $\delta^{15}$N of mussels at MC (and therefore the negative relationship with TDIN) could be due to the fact that at higher nitrogen concentrations can lead to primary producers being choosier which leads to a negative relationship between nutrient concentration and mussel (Page 12 Line 20 – Page 13 Line 10). | Page 12 Line 20 |
| | | We further explore the idea that MC is different by adding the following sentence:
" Site MC was closest to the ocean, was one of the deepest sites and had a higher TDIN concentration compared to all other sites, which in turn did not show differences in TDIN concentrations between them. This emphasises that the differences in mussel $\delta^{15}$N between sites detected in our estuary might rather reflect site-specific nutrient cycling processes than nitrogen pollution itself." | Page 13 Line 21 |
| | | In addition, we include the reviewer's idea that POM is originating from outside the estuary (upstream). This is a very interesting speculation and we added this into the discussion as follows:
"An alternative explanation would be that POM could originate upstream where nitrate might have had higher $\delta^{15}$N values (not quantified in this study). Upon entering the estuary, POM mixes with estuarine POM, uncoupling the within-estuary $\delta^{15}$N nitrate and POM $\delta^{15}$N values. This could also explain the strong relationship between $\delta^{15}$N in | Page 13 Line 2 |

| | | mussels and the distance from the estuary mouth found in our study. Such a strong relationship can be expected in estuaries with low pollution levels due to the aforementioned mixing, while little spatial variability in $\delta^{15}N$ values of primary consumers can be expected in heavily polluted estuaries due to the dominance of upstream POM, as was shown by Oczkowski et al. (2008)." | |
|---|---|---|---|
| 46 | Page 13
18 but your nitrogen sources of nitrate and ammonium were not different between sites (except MC) so this seems unlikely to explain differences in mussel 15N, no? More likely differences in POM 15N drove differences in mussel 15N and is reflected in relationship between mussel 15N and distance from mouth. It seems like there are other n cycling effects that are occurring here and could help to explain the negative (or lack of) correlation between 15N-NO3 and 15N-mussel (or TDIN and 15N-mussel. | We agree that differences in POM $^{15}N$ might drive differences in mussel $^{15}N$ and that this could be reflected in relationship between mussel $^{15}N$ and distance from the estuary mouth. We therefore deleted this section and incorporated parts of it earlier within the discussion, specifically where we now discuss the strong relationship between mussel $^{15}N$ and distance to estuary mouth:
"An alternative explanation would be that POM could originate upstream where nitrate might have had higher $\delta^{15}N$ values (not quantified in this study). Upon entering the estuary, POM mixes with estuarine POM, uncoupling the within-estuary $\delta^{15}N$ nitrate and POM $\delta^{15}N$ values. This could also explain the strong relationship between $\delta^{15}N$ in mussels and the distance from the estuary mouth found in our study. Such a strong relationship can be expected in estuaries with low pollution levels due to the aforementioned mixing, while little spatial variability in $\delta^{15}N$ values of primary consumers can be expected in heavily polluted estuaries due to the dominance of upstream POM, as was shown by Oczkowski et al. (2008)." | Page 13 Line 2 |

| 47 | Page 13
18 Your MC site may be influencing interpretation too much. If you had to interpret these data without the MC site, how would you do so? Does it change your overall conclusions? | We agree and weakened the dependency of the discussion using relationships only. We will do this by

i) Acknowledging that the relationships are strongly driven by MC (see also our reply to your comment #38 & #43):
"The concentrations of total dissolved inorganic nitrogen …. were higher towards the estuary mouth (Fig. 2), although these relationships were weak …. and were driven by site MC only."

"In the Swan River estuary, $NO_3$ was enriched and there was a positive relationship between nitrate $\delta^{15}N$ and the concentration of $NO_x$ throughout the estuary, although this was strongly driven by site MC."

"We also found a positive relationship between food (POM) and mussel $\delta^{15}N$, but a negative relationship between nitrate $\delta^{15}N$ and consumers (mussels), which was strongly affected by site MC."

"The relationship between mussel $\delta^{15}N$ and TDIN concentration within the estuary was much stronger when omitting site Cl and not significant when omitting site MC."

ii) Interpreting the data without site MC:" " Site MC was closest to the ocean, was one of the deepest |

Page 9 Line 5

Page 11 Line 15

Page 12 Line 20

Page 13 Line 11

Page 13 Line 21 |

| | | sites and had a higher TDIN concentration compared to all other sites, which in turn did not show differences in TDIN concentrations between them. This emphasises that the differences in mussel $\delta^{15}$N between sites detected in our estuary might rather reflect site-specific nutrient cycling processes than nitrogen pollution itself." | |
| | | This will not change our overall conclusion that mussels can be used as indicators for site-specific differences in pollution or nutrient cycling, which is "…important information for local management, but would have gone undetected at high pollution levels as the larger deviations of nitrogen stable isotope values would have made such small differences in mussel values invisible." | Page 14 Line 10, 27 |
| 48 | Page 13 18 I would like to see more results regarding the POM and its connection to N-cycling. You have a fairly strong trend between 15N-mussel and distance from estuary mouth. What is driving this? | i) We agree and added additional results for POM in the results section (3.4. Particulate organic matter (POM) $\delta^{15}$N values). This now includes the following information: "POM $\delta^{15}$N values were between 6.2 and 9.9 ‰ with no significant difference between sites ($F_{6,25}= 1.327$). A weak but significant negative relationship between POM $\delta^{15}$N values and TDIN concentration was detected ($r^2 =0.163$, $y = -0.044x + 9.37$, $F_{1,28} = 5.44$), while a significant positive relationship between nitrogen stable isotope signatures of POM and mussels was found ($r^2 =0.303$, $y = 0.20x + 7.40$, $F_{1,14} = 6.08$) (Fig. 4). The relationship between $\delta^{15}$N of POM and nitrate was not significant; however as this calculation was based on only five data points | i) Page 9 Line 27 |

| | | | |
|---|---|---|---|
| | | where simultaneous measurements of the two $\delta^{15}N$ values were available, the value of this result is uncertain." | |
| | | ii) We also included an additional figure showing the positive relationship between mussel and POM $\delta^{15}N$ (Figure 4) (please also see your comment #22) | ii) Figure 4; Reference in text on Page 10 Line 1 |
| | | iii) We now also discuss the strong trend between $^{15}N$ of mussel and distance from estuary mouth as follows:"…. "An alternative explanation would be that POM could originate upstream where nitrate might have had higher $\delta^{15}N$ values (not quantified in this study). Upon entering the estuary, POM mixes with estuarine POM, uncoupling the within-estuary $\delta^{15}N$ nitrate and POM $\delta^{15}N$ values. This could also explain the strong relationship between $\delta^{15}N$ in mussels and the distance from the estuary mouth found in our study. Such a strong relationship can be expected in estuaries with low pollution levels due to the aforementioned mixing, while little spatial variability in $\delta^{15}N$ values of primary consumers can be expected in heavily polluted estuaries due to the dominance of upstream POM, as was shown by Oczkowski et al. (2008)." | iii) Page 13 Line 2 |
| 49 | Page 14
- 4 correlated to nitrogen concentrations. But these were all negative correlations, no? | We agree that this sentence was misleading. To avoid this we deleted "…that correlated to differences in nitrogen concentrations...". | Page 14 Line 21 |
| 50 | Page 14
15N-mussel negatively correlated to 15N-NO3. 15N- | Because there are stable differences in mussel $\delta 15N$ between sites, we like to argue that mussels are | |

| | | | |
|---|---|---|---|
| | NO3 positively (though MC weighs heavily) related to NOx concentration. High NO3 reflected in 15N-NO3 does not appear in 15N-mussels as sites with high 15N-NO3 and NO3 have low 15N-mussels, no? So mussels don't appear to be good indicators of NO3 sources as they don't reflect 15N-NO3, no? | good indicators for site specific nutrient cycling, although we agree that in our system mussels are not good indicators for $NO_3$ sources themselves. We believe that this is due to the fact that this estuary showed low nitrogen pollution during the study period (e.g., Page 11 Line 28; Page 12 Line 19). Please also see our reply to your comment #5.

To reflect what we have just said, we rewrote this section as follows:
"The stable spatial differences in mussel $\delta^{15}N$ values over time highlight the value of this organism as a bioindicator of spatial water quality assessment. The negative trends between mussel $\delta^{15}N$ values and nitrate concentration or nitrate $\delta^{15}N$ values emphasize that mussels might not be good indicators for $NO_3$ sources in systems with low pollution levels. Instead, the small differences in mussel stable isotope signatures might reflect differences in site specific nutrient cycling caused by physicochemical conditions or biological factors rather than nitrogen pollution." | Page 14 Line 20 |
| 51 | Page14:
In the discussion and conclusion sections you refer to mussels reflecting nitrate pollution but the link is weak, dependent on MC, and negative. Explain how these connections interact or simplify your message in the discussion and conclusion. The emphasis appears to be on nitrate but the linkages between nitrate and mussel tissue are unclear. | We agree with this comment and simplified the conclusion to:
"The stable spatial differences in mussel $\delta^{15}N$ values over time highlight the value of this organism as a bioindicator of spatial water quality assessment. The negative trends between mussel $\delta^{15}N$ values and nitrate concentration or nitrate $\delta^{15}N$ values emphasize that mussels might not be good indicators for $NO_3$ sources in systems with low | Page 14 Line 20 |

| | | | |
|---|---|---|---|
| | | pollution levels. Instead, the small differences in mussel stable isotope signatures might reflect differences in site specific nutrient cycling caused by physicochemical conditions or biological factors rather than nitrogen pollution." | |

| | **Reviewer 2** | | |
|---|---|---|---|
| | **Comments** | **Response** | **Location in text** |
| 1 | [Structure]
The article is well written and structured. When going first through the manuscript, I had the impression that the introduction was pretty long (it is almost 0.25% of the text). Having said that, there is a lot of useful information and references included. One option could be to shorten a bit the introduction, or to introduce a few sub-headings in order to make it an easier read: basically it is about (1) increasing human impact on aquatic ecosystems, (2) the need for a better understanding of the spatial and temporal variability of pollution levels with a view to better manage these often irreversibly impacted systems, (3) the focus on nutrient pollution, (4) the use of stable isotopes (especially of N) for investigating anthropogenic nutrient pollution, and (5) the introduction of mussels as a sentinel organism in that specific context. | We agree that the introduction was too long and have shortened and streamlined it by,

i) deleting the third sentence of the first paragraph ("The high percentage….).

ii) deleting the last two sentences of the first paragraph ("In an attempt to reconnect….").

iii) deleting the first sentence of the second paragraph to make sure that we get to the point more quickly ("Typically the success rate…")

iv) delete part of the third paragraph ("and will be even more impacted in the future. Nutrient pollution is of particular concern in many waterbodies, because…")

v) being more specific about pollution (in the third paragraph) by exchanging the words "nutrient" and "pollutant" with "nitrogen". |

i) Page 2 Line 9

ii) Page 2 Line 14-23

iii) Page 2 Line 15

iv) Page 2 Line 25

v) e.g., Page 1 Line 22; Page 2 Lines 26, 28, 30, 31, 32; Page 3 lines 1, 3, 7; |

| | | vi) including a paragraph on our study organisms (*Mytilus edulis*, blue mussel) and its use as an indicator species.

By this, we believe that we have achieves a good balance between "getting to the point quickly" and "giving a broad picture of state of pollution management", which we think is appropriate for HESS that has such a large community of readers. | vi) Page 3 Line 33 – Page 4 Line 12 |
|---|---|---|---|
| 2 | 1-Introduction [pages 2-4]: When reading the introduction, and more specifically the paragraphs to the end where mussels are introduce as sentinel organisms, I was surprised (unless I am mistaken) not to learn about what species have eventually been used for this study. I think this is a very important aspect that the authors have not taken into consideration for their manuscript. In an area where they expect living organisms to be a living archive of the local average environmental conditions it is essential to know a minimum about the metabolism of that organism. Especially in a journal that has a large community of readers from hydrological sciences, we cannot necessarily expect them to know much about this topic. Moreover, since this is a kind of proof-of-concept study, the authors should carefully describe the organisms, growth rates, sensitivity to changing environmental conditions etc. These aspects are likely to be crucial when it comes to eventually understand and discuss the isotopic signatures of N in the mussel's foot tissue. As mentioned further down in this assessment, there is existing literature in this respect | We agree and now specified the species that we used (i.e. blue mussel, *Mytilus edulis*) throughout the manuscript. The sentences will now read as follows:

Abstract: "The main aim of this study was to assess the suitability of nitrogen stable isotope as measured in mussels (*Mytilus edulis*), as an indicator able to resolve spatial and temporal variability of nitrogen pollution in an urban, tidally influenced estuary (Swan River estuary; Western Australia)."

Introduction: "Bivalves on the other hand, which include the blue mussel are primary consumers with limited movement, and have been suggested as suitable site-specific bioindicators of time-averaged persistence of nutrient pollutants, because their isotopic signature fluctuates less than that of their food sources due to longer tissue turnover rates (Raikow and Hamilton, 2001; Post, 2002; Fukumori et al., 2008; Fertig et al., 2010; Wang et al., 2013)." | Page 1 Line 20

Page 3 Line 33 |

| | and it would certainly be of value to take this into consideration in a revised version of the manuscript. | Introduction: "The main aim of this study was to identify the variability of nitrogen concentration in an urban estuary over time and space and to ascertain the suitability of the isotopic signature ($\delta^{15}$N) of blue mussel (*Mytlius edulis*) tissue as an indicator of nitrogen pollution in urban water systems." | Page 3 Line 20 |
| | | Materials and methods: "Seven sites within the Lower Swan River estuary were sampled 6 times for blue mussels and 9 times for nutrients,…" | Page 5 Line 19 |
| | | Materials and Methods: "Nine blue mussels per site were randomly taken from the pylons of the jetties at each site from between 20 and 40 cm depth and brought into the laboratory on ice in bags containing water from the respective site." | Page 6 Line 10 |
| | | We further included a short paragraph to introduce this mussel species in the introduction: "The blue mussel, *Mytilus edulis*, is a common sessile bivalve in estuarine and marine environments that is able to adapt to a wide range of environmental conditions, such as food concentration, temperature and salinity (e.g., Thompson and Bayne, 1974; Widdows et al., 1979; Zandee et al., 1980; Almadavillela, 1984), and that shows low sensitivity to anthropogenic pressures (Mainwaring et al., 2014).  As such, this species is able to thrive at different pollution levels and has | Page 4 Line 5-12 |

| | | therefore been used as an indicator species for pollution (Phillips, 1976) and as a model organism for physiological, genetic and toxicological studies (Luedeking and Koehler, 2004) for some time." | |
|---|---|---|---|
| 3 | 2-Material and methods [page 5 study sites & 6 sampling and analyses]: When reading the changing conditions in the Swan River estuary, one could expect differences between mussel species that are exposed to these fluctuations in salinity (between high tide and low tide). Is there only one mussel species in the studied area? If not (what is very likely), what are the other species that are present – what species has the sampling protocol been targeting – was it a mix of species – how sure can we be that different sensitivities to changing environmental conditions (including pollution) can lead to differences in metabolic activity? | We believe that this comment directly links to your previous comment #2 and by clarifying that we only used one species (blue mussel, *Mytilus edulis*) we believe this comment has been addressed by our previous reply. We would like to note that by using only one species we made sure that differences in metabolisms are restricted to within-species variability. | i.e. Page 1 Line 20, Page 3 Line 33, Page 3 Line 20, Page 5 Line 19, Page 6 Line 10, Page 4 Line 5 |
| 4 | 3-Results [page 8 physicochemical parameters]: given that the study was carried out during rather dry conditions, the prevailing environmental parameters measured in the investigated area have also been rather unusual as stated in the manuscript. Here again, it would be interesting to see how the mussels populations have responded to that (if at all) – is there any information available on that? | We agree that conditions were unusually dry during our study. Unfortunately there are no previous data on this mussel population (e.g., abundance, physiology) that could be used for comparison with our study.

We would like to emphasise that blue mussels are known to adapt well to varying conditions. We have stated this now in the revised manuscript version. "The blue mussel, *Mytilus edulis*, is a common sessile bivalve in estuarine and marine environments that is able to adapt to a wide range of environmental conditions, such as food concentration, temperature and salinity (e.g., Thompson and Bayne, 1974; Widdows et al., 1979; | Page 4 Line 5 |

| | | | |
|---|---|---|---|
| | | Zandee et al., 1980; Almadavillela, 1984), and that shows low sensitivity to anthropogenic pressures (Mainwaring et al., 2014). As such, this species is able to thrive at different pollution levels and has therefore been used as an indicator species for pollution (Phillips, 1976) and as a model organism for physiological, genetic and toxicological studies (Luedeking and Koehler, 2004) for some time."

In addition, because the mussels within the estuary all experienced the same conditions, the dry conditions will not affect our conclusions. | |
| 5 | 3-Results [page 8 physicochemical parameters]: On page 8, line 10 units should be added to salinity. | We would like to note that salinity does not have a unit as it is a ratio of the conductivity of a seawater sample and a standard potassium chloride solution (see UNESCO (1985): The international system of units (SI) in oceanography, UNESCO Technical Papers No. 45, IAPSO Pub. Sci. No. 32, Paris, France.) We therefore did not include a unit. | - |
| 6 | 3-Results [page 8 physicochemical parameters]: On page 10 the delta symbol should be homogenised. | We agree and homogenised delta symbols by avoiding using them as a capital symbol at the beginning of a sentence. | Page 10 Line 6 and 8 |
| 7 | 4-Discussion [page 13]: In lines 6 to 8 I would be careful when stating that stable isotope signatures in mussels of tidally influenced estuaries are less impacted by seasonal changes in watershed input and chemistry compared to large rivers. This statement make sense considering the results of this study, but given the particularly dry conditions that prevailed during this investigation and the proof-of-concept character of this study, there need most probably to be more | We agree with this and have weakened this statement by restating it as follows:
"Our results therefore highlight that while high seasonal variations of stable isotope signature in mussels can be connected to seasonal changes in watershed input and chemistry in large rivers (Fry and Allen, 2003), this is less pronounced in tidally influenced estuaries or during drier conditions with low freshwater input." | Page 14 Line 2 |

| | | | |
|---|---|---|---|
| | investigations before a strong statement in this sense. | | |
| 8 | 5-Conclusion [page 13]: A similar comment as for the point above can be made for the 1st paragraph of the conclusion. | We agree and rewrote the second sentence of the first paragraph as: "As such, stable isotope analysis of a model organism, such as the blue mussel can deliver essential information for future decentralised water management practices that are focused on local process understanding." | Page 14 Line 10 |
| 9 | 5-Conclusion [page 13]: Of interest could also be to see if there are differences in signatures between species. | We only analysed stable isotope signature of one species, blue mussel, which we now clarified throughout the manuscript as shown in our reply to your comments #2 and #3. | i.e. Page 1 Line 20, Page 3 Line 33, Page 3 Line 20, Page 5 Line 19, Page 6 Line 10, Page 4 Line 5 |
| 10 | 5-Conclusion [page 13]: In the conclusion it is stated that the future studies should contribute in similar (low) polluted systems to better understand the baseline of spatial natural isotopic variability in urban aquatic systems. I was wondering if this is not somehow contradictory with what is announced in the title – are mussels then really used in the sense of sentinels of pollution or rather as indicators of the baseline of 'spatial natural isotopic variability in urban aquatic systems'.

Again here I am possibly confused by the fact that no | We agree that this was misleading and we therefore restated this sentence. This now presents an additional suggestion for future studies to gain a better understanding of systems with varying and partly low pollution levels. We rewrote it as follows: "In addition, we advocate future studies in similarly (low) polluted systems that include stable isotope analysis of other food web end-members and nutrients of the groundwater, to develop baselines of spatial natural isotopic variability in urban aquatic systems which will help identifying the importance of local biogeochemical processes for pollution control." We believe that this is reflected in the title.

We agree that this information has been missing | Page 14 Line 30 |

| | | | |
|---|---|---|---|
| | information is given on how sensitive those organisms are eventually to pollution. | and have now included that blue mussels are not very sensitive to pollution by human activities. As such, this organism is able to thrive at different pollution levels indicating that their stable isotope signature should be an ideal indicator to identify differences in pollution levels. To reflect what we have just said, we included the following paragraph: "The blue mussel, *Mytilus edulis*, is a common sessile bivalve in estuarine and marine environments that is able to adapt to a wide range of environmental conditions, such as food concentration, temperature and salinity (e.g., Thompson and Bayne, 1974; Widdows et al., 1979; Zandee et al., 1980; Almadavillela, 1984), and that shows low sensitivity to anthropogenic pressures (Mainwaring et al., 2014). As such, this species is able to thrive at different pollution levels and has therefore been used as an indicator species for pollution (Phillips, 1976) and as a model organism for physiological, genetic and toxicological studies (Luedeking and Koehler, 2004) for some time." | Page 4 Line 5 |
| 11 | 5-Conclusion [page 13]: As a last comment, one could also say that nutrient pollution is not really an urban problem or at least the origin of it can most of the time be found further upstream in agricultural parts of the catchments. In urban environments, one could also be targeting other sources of pollution, such as heavy metals, xenobiotics, etc. | We agree with this and we now mention its future application as sentinels for non-nutrient co-occurring pollutants (such as oils, heavy metals) in

i) the abstract:" We suggest that mussels and other sentinel organisms can become a robust tool for the detection and characterization of the dynamics of a number of emerging anthropogenic pollutants of concern in urban water systems." | i) Page 2 Line 1 |

| | | | |
|---|---|---|---|
| | | ii) the conclusion: "We propose to further investigate its use for assessing the pollution by co-occurring non-nutrient pollutants, such as oils and heavy metals, which are entering waterbodies simultaneously with nutrients during stormwater events and which are critical in urban systems." | ii) Page 14 Line 12 |
| 12 | Concluding remarks: This manuscript is certainly a very interesting contribution for the readers of this journal and I enjoyed very much reading it. It is an interesting case study – or more specifically a proof-of-concept study – introducing mussels as a sentinel organism for investigating nutrient pollution in an urban aquatic environment. Since existing literature on similar applications/studies is not much referred to in the manuscript, the innovative character of this study might however be slightly overrated. | We agree that using mussels as an indicator for pollution is not new and we now included more references (Wang et al. 2013; Wen et al 2010; Fry et al 2011) that looked at mussels as indicators of nutrient pollution in lakes and estuaries: "In addition and identical to our study, the range of $\delta^{15}N$ values for nitrate and POM has been shown to be wider than the range for primary producers, indicating a time-averaging effect in mussels (Gustafson et al., 2007; Wang et al., 2013). Previous studies reported mussel $\delta^{15}N$ values between +6.6 and +16.7 ‰ in densely populated areas (Cabana and Rasmussen, 1996), polluted inland waterbodies (Wen et al., 2010; Wang et al., 2013) and a eutrophic estuary (Fry et al., 2011)." | Page 12 Line 12 |
| | | We have further added a recent publication on the use of other primary producers (non-mussels) as indicators of nutrient pollution to show the wide use of this approach (Xu and Zhang 2012). | Page 3 Line 17 |
| | | We further agree with you that the use of this approach in an urban context makes this study novel and interesting. We now highlighted this | |

| | | within the | |
|---|---|---|---|
| | | - abstract:"We suggest that mussels and other sentinel organisms can become a robust tool for the detection and characterization of the dynamics of a number of emerging anthropogenic pollutants of concern in urban water systems." | Page 2 Line 1 |
| | | - introduction: "However, very little information exists on the use of these stable isotopic signatures in urban systems." | Page 4 Line 18 |
| | | -conclusion: "With an increasing importance of managing urban aquatic systems sustainably, our work presents an important proof-of concept study in this context. " | Page 15 Line 5 |